# Illness-Promoting Psychological Processes in Children and Adolescents with Functional Neurological Disorder

**DOI:** 10.3390/children10111724

**Published:** 2023-10-24

**Authors:** Kasia Kozlowska, Olivia Schollar-Root, Blanche Savage, Clare Hawkes, Catherine Chudleigh, Jyoti Raghunandan, Stephen Scher, Helene Helgeland

**Affiliations:** 1Department of Psychological Medicine, The Children’s Hospital at Westmead, Westmead, NSW 2145, Australia; olivia.schollarroot@health.nsw.gov.au (O.S.-R.); blanche@goldenwattleclinicalpsychology.com.au (B.S.); clare.hawkes@health.nsw.gov.au (C.H.); jyoti.raghunandan@health.nsw.gov.au (J.R.); 2Child and Adolescent Heath and Specialty of Psychiatry, Faculty of Medicine and Health, University of Sydney, Camperdown, NSW 2050, Australia; 3Brain Dynamics Centre, Westmead Institute of Medical Research, Faculty of Medicine and Health, University of Sydney, Westmead, NSW 2145, Australia; 4Golden Wattle Clinical Psychology, 20 Jarrett St, Leichhardt, NSW 2040, Australia; 5Department of Psychiatry, McLean Hospital, Harvard Medical School, Boston, MA 02115, USA; sscher@mclean.harvard.edu; 6Specialty of Psychiatry, Faculty of Medicine and Health, University of Sydney, Camperdown, NSW 2050, Australia; 7Department of Child and Adolescent Mental Health in Hospitals, Oslo University Hospital, 0424 Oslo, Norway; helene.helgeland@ous-hf.no

**Keywords:** biopsychosocial, cognitive behavioural therapy (CBT), functional neurological (conversion) disorder (FND), functional seizures, hypnosis, neural network dysregulation, psychological, psychotherapy, retraining and control therapy (ReACT), stress-system activation, top-down regulation strategies, treatment

## Abstract

Previous studies suggest that subjective distress in children with functional neurological disorder (FND) is associated with stress-system dysregulation and modulates aberrant changes in neural networks. The current study documents illness-promoting psychological processes in 76 children with FND (60 girls and 16 boys, aged 10.00−17.08 years) admitted to the Mind–Body Program. The children completed a comprehensive family assessment and self-report measures, and they worked with the clinical team to identify psychological processes during their inpatient admission. A total of 47 healthy controls (35 girls and 12 boys, aged 8.58–17.92 years) also completed self-report measures, but were not assessed for illness-promoting psychological processes. Children with FND (vs. controls) reported higher levels of subjective distress (total DASS score, *t*(104.24) = 12.18; *p* ˂ 0.001) and more adverse childhood experiences across their lifespans (total ELSQ score, *t*(88.57) = 9.38; *p* ˂ 0.001). Illness-promoting psychological processes were identified in all children with FND. Most common were the following: chronic worries about schoolwork, friendships, or parental wellbeing (*n* = 64; 84.2%); attention to symptoms (*n* = 61; 80.3%); feeling sad (*n* = 58; 76.3%); experiencing a low sense of control (helplessness) in relation to symptoms (*n* = 44; 57.9%); pushing difficult thoughts out of mind (*n* = 44; 57.9%); self-critical rumination (*n* = 42; 55.3%); negative/catastrophic-symptom expectations (*n* = 40; 52.6%); avoidance of activities (*n* = 38; 50%); intrusive thoughts/feelings/memories associated with adverse events (*n* = 38, 50%); and pushing difficult feelings out of mind (*n* = 37; 48.7%). In children with FND—disabled enough to be admitted for inpatient treatment—illness-promoting psychological processes are part of the clinical presentation. They contribute to the child’s ongoing sense of subjective distress, and if not addressed can maintain the illness process. A range of clinical interventions used to address illness-promoting psychological processes are discussed, along with illustrative vignettes.

## 1. Introduction

Functional neurological disorder (FND) is a mind–body disorder involving aberrant changes within neural (brain) networks as well as complex interactions between brain, mind, body, and context—the lived experience of the child and the family. Children (including adolescents) with FND present with a broad range of functional neurological symptoms (see Figure 1). In the hospital setting, presentations with FND are common [1,2,3,4]. During the last few decades, children in our society have become more stressed in the context of climate change, cumulative natural disasters, the COVID-19 pandemic [5,6], and increasing academic pressures, political tensions, and social stressors [7]; presentations with functional somatic symptoms have increased around the globe [8,9,10,11,12,13,14,15,16,17,18]. In our own clinical contexts—tertiary care hospitals in Sydney (Australia) and Oslo (Norway)—the authors use a stress-system model to inform their work with children with functional somatic symptoms (including FND) [19,20].

The stress-system model is a biopsychosocial (systems) framework for looking at the diverse, and complex interrelated biological, psychological, family, social, and ecological systems that underpin FND and other stress-related illnesses (Figure 2 and Figure 3) [19,20]. In the current study we examine the psychological—*mind* system level—which encompasses the mental categories of thinking (cognition) and feeling. In psychology, thinking is defined as a “cognitive behavior in which ideas, images, mental representations, or other hypothetical elements of thought are experienced or manipulated” [21]. In this sense, feelings, the mental representations of body states (emotions)—including the feeling of physiological states such as thirst, pain, and fatigue [22]—reflect a particular subset of cognitions. Feeling-related mental representations take the form of words or images, and they allow for subjective awareness, the felt sense of the body. Thoughts and emotions (body states) are linked in a rich reciprocal network, and “certain thoughts evoke certain emotions and vice versa” (p. 71) [23]. We look at illness-promoting psychological processes that emerge in the minds of children with FND and that function—in a top-down fashion—to activate and modulate the stress system, thereby contributing to the activation or maintenance of the illness process [23,24,25,26,27]. Because girls with FND present more frequently than boys in civilian settings [4,28,29,30,31,32,33,34,35], throughout this article, when using pronouns, we generally use the pronoun *she*.

The stress system is a key player in the neurobiology of FND (see Figure 2) [37,38,39,40,41,42,43,44]. The stress system maintains energy regulation (allostasis) [45,46], regulates the physiological state of the body (milieu intérieur) [47], and modulates the brain–body stress system in response to threat and the challenges of daily living (see Figure 3) [48]. In health and wellbeing, energy regulation is efficient; the biological systems that underpin the physiological state of the body work in synchrony [49,50] and the stress system activates to meet the challenges of daily living and then returns to baseline function [19,45]. In a parallel process the brain “consolidate[s] the details of experience within the brain’s synaptic connections, making those experiences available to guide later decisions—usually unconsciously—about future challenges” (pp. 3–5) [26]. Some of these representations, however, are made available to consciousness in the form of thoughts, feelings, and mental images. Because of these predictive functions the brain is conceptualized as a predictive organ capable of generating representations that are used to help guide behaviour and to maintain allostasis [26,46].

In illness states—including FND—illness-promoting thoughts and feelings generated by the brain, as well as focus of attention (e.g., on symptoms), can be maladaptive. These psychological processes can have an illness-promoting top-down effect on stress-system activation or dysregulation, and they can function as drivers of the illness process: FND symptoms, pain, sleep disturbance, comorbid nonspecific functional somatic symptoms, and illness-promoting behaviours [25,38,39]. FND symptoms increase when the child or family place their attention on the symptoms, and they decrease or disappear when the child’s attention (or that of the family) is directed elsewhere [19,51]. This aspect of FND is utilized by neurologists during the neurology examination to demonstrate variability of symptoms with distraction and across contexts [52,53,54,55]. Taken together, this body of work suggests that the mind and the body are parts of an integrated whole—a biopsychological system embedded in a socio-ecological context (see Figure 3). It is therefore not surprising that for a large majority of children—and their families—working with focus of attention and illness-promoting thoughts, feelings, and behaviours is a key component of the treatment intervention.

In the research literature, a small number of studies have examined illness-promoting psychological processes in children with FND. We discuss them briefly.

Assessments of attachment using the dynamic-maturational model of attachment and adaptation (DMM) classify children with FND (vs. healthy controls) into at-risk patterns of attachment with high rates of unresolved loss and trauma [56]. At-risk patterns of attachment are characterized by distortions in information processing that can be identified via linguistic analysis of interview material across five memory systems (procedural, imaged, semantic, episodic, working memory). The coder assigns the interview a pattern of attachment that best describes the organization of the child’s interpersonal emotional and cognitive responses to perceived threat. Patterns of attachment fall into three clusters: a normative cluster and two “at-risk” clusters—the A+ cluster and the coercive C+ cluster (see Figure 4). The coder also identifies discourse markers of unresolved loss or trauma—dangerous or distressing past events that the speaker struggles to integrate into the narrative in a coherent way [57,58,59,60,61]. In a study with children disabled by FND (*n* = 76), patients were classified into the Type A+ and Type C+ clusters, whereas controls mostly classified into the normative cluster [56]. Children with FND also showed higher rates of unresolved loss or trauma than healthy controls (75% vs. 12%) (see Figure 4).

Personality testing using the Millon adolescent clinical inventory (MACI) with children (*n* = 20) with functional seizures (an FND subtype; see Figure 1) showed high levels of psychic tension, anxiety, depressive affect, and maladaptive personality traits, suggesting maladaptive patterns of coping [62,63]. In a different study using the MACI with children with functional seizures (*n* = 42), no differences between patients and controls were found [64]. The latter study also found no differences between patients and controls on adolescent psychiatric and emotional functioning (using the Behaviour Assessment System for Children, Second Edition), childhood trauma (using the Childhood Trauma Questionnaire, a self-report measure of physical, sexual, and emotional abuse and physical and emotional neglect), and body awareness (using the Body Awareness Multidimensional Assessment of Interoceptive Awareness). The study did find that children with functional seizures had greater somatization (using the Children’s Somatic Symptoms Inventory–24) and reported a poorer quality of life (using the Paediatric Quality of Life Inventory Generic Core Scales).

The lack of findings on measures validated for other populations—anxiety, depression, and externalizing disorders—is consistent with previous findings pertaining to the lack of utility of many self-report measures in children with FND. One study found that even in children with FND (*n* = 57) who were admitted to hospital with mixed somatic and psychological symptoms and high levels of functional disability, self-report measures commonly used by mental health clinicians did not identify these children as distressed and as requiring clinical intervention [65]. Our clinical experience suggests that many children with FND do not resonate with *psychological constructs* used by mental health clinicians. For example, if the clinician probes for hyperarousal symptoms using anxiety-related constructs—for example, “did you feel anxious?”—the response is likely to be negative. Alternately, if the clinician probes for hyperarousal symptoms using physical constructs—“did you experience fast breathing, flushing, going hot and cold, butterflies in the stomach, nausea, vomiting, and difficulties thinking clearly?”—the response is likely to be positive. The implication is that the diagnosis of comorbid mental health concerns in children with FND is much more likely to be accurate when it is based on clinical interview rather than self-report measures using psychological constructs [32,65].

The broader literature also highlights the importance of using a broad-lens approach to assess for stress and adverse childhood experiences (ACEs). In contrast to adult studies, paediatric studies generally show that rates of maltreatment are relatively low. Instead, the more common pattern of presentation is that of cumulative common-place stressors that exceed the child’s adaptive capacities to cope [38,56,66], Such stressors include physical stressors (illness, injury, and medical procedures), relational stressors (family stress, parental psychiatric illness [31,32], friendship problems, bullying, and social exclusion), academic stress (a difficult transition to high school, high academic expectations exceeding the child’s capacity), high sporting expectations from the self or others (a relentless training schedule), and more recently, exposure to increasing natural disasters and social stress associated with the pandemic [4,18,38]. Importantly, illness-promoting focus of attention and illness-promoting thoughts and feelings—often related to the ACEs experienced by the child—contribute to stress-system activation.

In addition, expectations regarding negative or catastrophic symptoms (known as “catastrophic symptom expectations”) and a low sense of control over symptoms have been identified as important factors for maintaining FND symptoms (including functional seizures) [67,68]. In a laboratory-based setting, Stager and colleagues used a modified Stroop Task and the Magic and Turbulence Task to show that children with functional seizures (*n* = 26) were characterized by a low sense of control over symptoms and by an increased attention to illness-related stimuli [69]. The researchers also explored targeting these elements as part of the treatment intervention. They used both a specially developed cognitive behavioural therapy (CBT) intervention—termed retraining and control therapy (ReACT)—to target catastrophic thinking and habit-reversal strategies to shift attention (in which the child focused on doing a physical response incompatible with the symptoms of functional seizures). The ReACT intervention led to a resolution of functional seizures [67,70,71]. Shifting focus of attention away from symptoms is also a key element of other interventions for functional seizures [20,62,63].

The current study documents the illness-promoting psychological processes that were identified—and became targets of treatment—in a cohort of children with FND who were admitted and ongoingly evaluated by a multidisciplinary team during a two-week mind–body hospital admission. We (the authors) also use this opportunity to briefly describe the treatment interventions themselves and to provide example case vignettes.

## 2. Methods

### 2.1. Participants

Eighty children admitted for treatment of FND to the inpatient Mind–Body Program at the Children’s Hospital at Westmead [72] during the period January 2019 to May 2023 agreed to participate in the FND research study. All children had undergone a comprehensive neurology assessment and had been given a Diagnostic and Statistical Manual of Mental Disorders (DSM-5) diagnosis of FND by a paediatric neurologist [73]. All had also participated in a biopsychosocial assessment with the mind–body team: a structured interview with the child and family documenting the child’s developmental history, history of the presenting symptoms (including comorbid, nonspecific symptoms), and functional disability rating on the Global Assessment of Functioning (GAF) scale. On self-report the children completed the Depression Anxiety and Stress Scales (DASS-21) and Early Life Stress Questionnaire (ELSQ) (See Table 1).

Forty-seven healthy controls had been recruited into the study from the same age bracket and geographical catchment area [38,39]. Control participants were screened for the absence of the following: mental health disorders, history of head injury, family history of mental health disorders, and chronic health concerns. All controls completed the DASS-21 and ELSQ and were rated with the GAF by a parent (see Table 1). Controls were not assessed for illness-promoting psychological processes.

The study was approved by the Sydney Children’s Hospital Network Ethics Committee (HREC/18/SCHN/232) on 7 August 2018. Participants and their legal guardians provided written informed consent. Three children and their families provided additional written consent enabling the mind–body team to use examples of their illness-promoting psychological processes in teaching and to provide concrete examples for this article. Other examples reflect amalgamation of similar cases (amalgam cases).

### 2.2. Data Acquisition

The identification and documentation of illness-promoting psychological processes took place during the two-week mind–body admission (the FND group only). Identification of illness-promoting psychological processes was a stated goal of each admission process. Based on our clinical work with a previous cohort of patients, the mind–body team had developed a comprehensive list of illness-promoting psychological processes [78]. The list was designed as a therapeutic check list that could help the members of our team hold in mind the multiple overlapping processes that, if present, would become a target of the treatment intervention.

During the two-week mind–body admission, the child’s clinical team—clinical psychologist, child and adolescent psychiatrist, clinical nurse consultant, and paediatric registrar—worked closely with the child (and family) to identify any illness-promoting psychological processes that were part of the child’s FND presentation. Illness-promoting psychological processes were commonly observed in interactions with the child on the ward, in the school, in adolescent group, in physiotherapy, or in individual psychotherapy sessions. The illness-promoting psychological processes were identified, documented in the clinical notes, and identified—as a problem area that required a targeted therapeutic intervention—in individual psychotherapy sessions with the child and also in family sessions with the child and family. When possible, interventions to target one or more of the processes were implemented during the mind–body inpatient admission (a component of a broader biopsychosocial intervention) [72]. The psychological processes that required ongoing intervention were subsequently documented in the child’s discharge report and handed over—both verbally and via the report—to the child’s psychotherapist in the community. In this way, the therapeutic work pertaining to the illness-promoting psychological processes begun during the inpatient admission was continued as part of the child’s ongoing psychotherapy in the community, serving as a key element in the child and family’s taking the Mind–Body Program home [72].

### 2.3. Analysis of Clinical Characteristics and Psychological Processes Data

Using SPSS Statistics 26, we performed descriptive statistics, chi-square analyses and independent *t*-tests to describe the clinical characteristics of the FND participants and to calculate differences between the FND and control groups on categorical and continuous variables, respectively.

### 2.4. Missing Data

Of the 80 children participating in the FND research study, data pertaining to illness-promoting psychological processes had not been documented for 4 children, leaving a cohort of 76. Of these 76, 4 had not completed the DASS and 3 had not completed the ELSQ. Reasons included being too ill on admission (*n* = 1), being too young to understand the questions on the DASS (*n* = 1) and failing to return both questionnaires (*n* = 2).

### 2.5. Participant Characteristics

The final study groups comprised 76 children (60 girls and 16 boys) with FND aged 10.00 to 17.08 years (mean = 13.75; SD = 1.74; median = 13.75) and 47 healthy controls (35 girls and 12 boys, aged 8.58 to 17.92 years [mean = 13.83; median = 14.33; SD = 5.93]). The groups were matched for sex (χ^2^ = 0.33; *p* = 0.565) and age (*t*(75.01) = −0.251; *p* = 0.802).

The clinical presentations of the 76 children with FND were diverse. They presented with one or more functional neurological symptoms (range = 1–8; mean = 3.13; median = 3.00; SD [standard deviation] = 1.90) (Figure 5). Length of illness ranged from 1 week to 4 years (mean = 5.91 months; median = 4.00 months; SD = 7.12 months); approximately two-thirds (50/76; 65.8%) had been ill for less than six months. Total number of bed days—including bed days for the medical workup and for longer-than-usual admissions—ranged from 5 to 133 days (mean = 23.05 days; median = 16.5 days; SD = 20.65). Levels of functional disability at clinical assessment were high, with GAF scores ranging from 10 to 51 (mean = 32.04; median = 31.00), with the score of 100 representing the highest possible score. Days of school loss ranged from 0 to 52 weeks (mean = 8.28; median = 4.50; SD = 10.03) on presentation. Clinical characteristics are reported in Table 2.

Eighty-seven percent (*n* = 66) had one or more comorbid, DSM-5 mental health disorders (range = 1–4, mean 1.59, median 2, SD 1.00) (see Table 2). Additionally, almost a third experienced suicidal ideation at presentation (*n* = 23; 30.3%).

Comorbid complex/chronic pain was present in approximately two thirds of children, other functional syndromes in almost a third, and a medical condition in just over a third (see Table 2).

In sum, the cohort reflected a very ill group of children with FND. Their functional impairment—meeting the threshold for an inpatient intervention—was due to the severity of their FND, coupled with functional impairment related to comorbid pain, other functional somatic symptoms (not meeting criteria for FND), and psychiatric comorbidities.

### 2.6. Self-Report Data and Global Assessment of Function

Relative to controls, patients with FND had significantly higher total scores on the DASS (subjective distress) and lower scores on the GAF (level of function). On the ELSQ they reported more ACEs across their lifespans (see Table 3). The ELSQ identified some additional ACEs or risk factors to that reported on clinical interview (see Table 4). More than a quarter of children with FND had been premature or experienced birth complications (*n* = 21; 27.6%). More than a quarter had experienced surgery or repeated hospitalization (*n* = 20; 26.3%). Over a quarter reported the experience of emotional abuse across family and other contexts (*n* = 20; 26.3%). And almost a fifth (*n* = 15; 19.7%) had witnessed a natural disaster first hand (see Table 4). Taken together this comparative analysis showed that our cohort of children with FND was characterised by a history of multiple ACEs, high levels of subjective distress, and significant impairment in function.

## 3. Results: Illness-Promoting Psychological Processes Identified during Admission to the Mind–Body Program (FND Group Only)

Illness-promoting psychological processes were identified for all children with FND (range = 1–24; mean = 12.31; median = 12.5; SD = 4.84) (see Table 5). The most common illness-promoting psychological processes pertained to the following domains:−Attentional processes: attention to symptoms (*n* = 61; 80.3%);−Cognitive processes and expectations related to the functional symptoms: e.g., negative/catastrophic-symptom expectations (*n* = 40; 52.6%), and experiencing a low sense of control (predictability) in relation to the symptoms (*n* = 44; 57.9%);−Cognitive processes (more general and related to the self): e.g., self-critical rumination (*n* = 42; 55.3%);−Feeling-related processes: e.g., feeling worried about schoolwork, friendships, or parental wellbeing (*n* = 64; 84.2), and feeling sad (*n* = 58; 76.3%);−Avoidance processes: e.g., pushing difficult thoughts out of mind (*n* = 44; 57.9%), pushing difficult feelings out of mind (feeling avoidance, *n* = 37; 48.7%), and avoiding activities (*n* = 38; 50.0%);−Disconnecting from the body and inability to sense and track body state (*n* = 43, 56.6%);−Unresolved loss, unresolved trauma, and unresolved bad experiences: intrusive thoughts/feelings/memories associated with the event (*n* = 38; 50.0%)

## 4. Treatment Interventions Targeting Illness-Promoting Psychological Processes

In this section we describe how the findings from our study translate into clinical practice. When working therapeutically with children with FND, a variety of approaches can be used to target illness-promoting psychological processes. We describe some of the interventions that we the authors—and our colleagues—use most frequently in our daily clinical practice.

The first set of interventions—setting up positive expectations, noticing the felt sense of the body (including changes in body state that emerge as body sensations), and targeting focus of attention—is centrally important and woven into all aspects of treatment. These interventions shape the treatment course and outcomes of all young people presenting with FND. Because of their importance, they are discussed in greater detail in the following section.

The second set of interventions—involving CBT, trauma processing, and hypnosis—constitute “blocks of psychological work” that may, when indicated, be integrated into the child’s individual psychotherapy treatment plan. This treatment plan runs alongside and includes the following: working with the family, reintegrating the child back to school, and (not covered in this article) mobilizing or increasing the child’s physical resilience in physiotherapy. This overarching (biopsychosocial) individualised treatment plan is informed by a biopsychosocial formulation that, following a comprehensive assessment, has been co-constructed with the child and family. The biopsychosocial assessment and process of co-constructing the formulation have been discussed elsewhere in detail [19,20,36,72,79,80].

It is also important to note that the psychological processes discussed in the coming sections are not unique to a diagnosis of FND. They occur across a range of stress-related and mental health disorders. As such, many of the psychological processes and interventions described below will be familiar to most mental health clinicians, and the interventions needed to address them will already be within the clinicians’ existing treatment “tool kit”. In this context, clinicians working with children with FND are encouraged to draw on their own skill sets and clinical expertise when developing a treatment plan. In addition, it is often the case that one treatment intervention may target several psychological processes concurrently. It is up to the treating clinician(s) to determine what interventions are required and when these should be delivered.

## 5. Centrally Important Interventions: Used with All Children and Their Families

### 5.1. Setting Up Positive Expectations

In the results section, we saw that illness-promoting cognitive processes and negative expectations pertaining to FND symptoms are common in children with FND. “Expectations shape—at least in part—the manner in which the brain processes a broad range of sensory, motor, interoceptive, and emotion-related information” (p. 86) [20]. The child’s expectations will thus be reflected in her body, thoughts, and feelings and can potentially, through a range of mind–brain–body pathways, affect the course of her symptoms [24,25,27,38,39]. Likewise, expectations that a child (and the family) hold toward treatment are a powerful contributor to the treatment process and to clinical outcomes [81]. In our interactions with the child and family—from the very first contact—we work to shape positive expectations with regard to the treatment process and the likelihood of good treatment outcomes. This process of setting up positive expectations needs to be embedded in the therapeutic relationship and will depend on the clinician’s capacity to connect with the child and family to create mutual trust and acceptance.

Vignette: Jack (use of an indirect positive suggestion)

Jack was a 13-year-old boy with functional gait difficulties. The physiotherapist began the process of removing Jack’s wheelchair, beginning with the footboard. She expressed the positive expectation that the wheelchair would soon be parked away—*when*, not if, Jack’s legs regained strength. Jack’s gait normalized four weeks later.

Vignette: Olga (use of a direct positive suggestion)

Olga, a 12-year-old girl experiencing functional seizures, attended an initial assessment with her distressed parents, who asked about her prognosis, “Will my child get well?” The team doctor explained that treatment outcomes are very positive. She said, “The research shows that children and adolescents do really well. Almost everyone—up to 95% of children—show full resolution of symptoms, especially when they work with a multidisciplinary team and engage in a holistic program: physiotherapy, individual therapy, working with the family, and working with the school. It will be interesting how you will notice this in your body.” Olga and her parents appeared relieved to hear this and expressed hope in the possibility of Olga’s recovery.

Vignette: Nancy (use of humour to shift expectations)

Nancy, a 13-year-old girl with limb weakness, tremors, visual symptoms, headaches, and dizziness, described her admission to her local hospital as traumatic. Her family described the staff’s clinical approach—at Nancy’s local hospital—as “punitive”. As a result, Nancy and the family had a lack of trust in the health system and were uncertain about an admission to the Mind–Body Program. During their assessment, clinicians listened attentively to the family’s story, providing space for them to discuss their difficult experiences. The clinicians also—deliberately—wove humour into the assessment in order to signal safety and play, not fault finding and sanctions. Together with the family, they laughed about the father’s extensive collection of horror figurines, thereby setting up expectations that the mind–body admission would have a light-heartened element and many opportunities for laughter.

Vignette: Meera (use of positive suggestion through hypnosis)

Meera was a 17-year-old adolescent with FND—paralysis of her legs and in one arm, with comorbid postural orthostatic tachycardia syndrome, fatigue, and social anxiety—that had not improved with treatment and that left her wheelchair-bound. On presentation Meera was hopeless and was clear that she did not expect anything to work. She did not expect to get better. To shift these expectations the therapist provided psychoeducation about the treatment outcomes for FND: she made explicit positive predictions about Meera’s likelihood of recovery. The therapists then focused on and amplified small gains that Meera was making, thereby redirecting Meera’s attention away from her perceived failures. The therapist then used hypnosis designed to build positive expectancies for the future (adapted from a Michael Yapko protocol for patients with depression). The work emphasized the idea that past (negative) experiences do not predict the future. Meera’s sense of agency and hope increased as she built the links between her actions today and her future outcome. The therapist used a mixture of general suggestions (e.g., “Every choice you make today will help to contribute to what tomorrow is like for you.”) and concrete suggestions (e.g., “Every time you choose to do your physio exercises you are helping yourself get better and better.”).

Vignette: James (use of direct positive suggestion and scaffolding)

James was an 11-year-old boy with functional seizures. He and his family had noticed a pattern where his functional seizures would subside during the school holidays and then return during school term, making it hard for James to attend school consistently. James and his therapist had worked on a variety of mind–body regulation strategies. During the school holidays James was able to use these strategies to avert a seizure when he noticed the impending warning signs. But he struggled to transfer these skills to the school environment. In a therapy appointment scheduled for the last week of the school term before the holidays, the therapist made the explicit prediction that he would do well in the school holidays, as he had always done so before. When James came for a review at the end of the holidays, he had (as expected) done very well (no functional seizures). The therapist then made a second prediction that, given how well he had done in the school holidays, he would be able to carry these skills through to the new school term. The session then focussed on what skills James had used in the school holidays and how he could carry these through to the school term (i.e., building up a sense of agency alongside positive expectations). At his next appointment, James was proud to report he had spent the last two weeks at school seizure free.

### 5.2. Working with the Felt Sense of the Body: Noticing and Sequencing Body Sensations

The felt sense of the body refers to the child’s capacity notice how her body is feeling and to notice changes in body state (body sensations). In children with FND, being able to notice states of activation that herald the onset of a functional episode—a functional seizure or a functional tic episode—are an important element of the therapeutic work. Felt sense of the body is a skill that originates from Eastern meditative traditions—including bodywork traditions—a form of bottom-up mindfulness [82,83,84,85]. Our mind–body team refers to this skill as noticing “the body’s level of activation or arousal” or noticing the “warning signs” of a functional seizure or tic episode (see Figure 6). From a neurophysiological perspective these “warning signs” reflect the neurophysiological activation or arousal that is a feature of paediatric FND and that precedes episodic symptoms (see Figure 7) [20].

Noticing these sensations—and sometimes sequencing them, so that the pattern of the episodes become “known”—is a skill in and of itself. The therapeutic work involves sensing the state of the body, followed by the task of tagging the body state with a word description (feeling words such as tense, painful, tight, icky, queasy, and so on) or image (visual representation). In our experience the creation of a visual representation of the felt sense of the body by the child and therapist—in the form of a body map drawing—can make the image (cognitive representation) very concrete and much easier to keep in mind. For some children the skill of noticing body sensations is difficult to attain; for these children it may take considerable practice.

Clinicians coming from the CBT framework also use this skill as part of their intervention. They may refer to the skill as simply “feeling an episode is about to begin” or as a type of mindfulness—a bottom-up mindfulness that involves awareness of the felt-sense of the body [82].

An important point about this skill is that, once the child has learnt to notice her “level of activation or arousal” or her “warning signs”, the next step of the intervention is to shift attention away from the symptoms or “the warning signs” and to implement active strategies to manage the symptoms (see following subsection).

### 5.3. Working with Focus of Attention

Illness-promoting attention to symptoms was identified in 80% of children with FND in the current study. When a child first becomes unwell, the physical symptoms signalled by the body bring attention to the fact that the child is sick and that a medical assessment is needed. In FND, however, once the medical assessment has been undertaken and a positive diagnosis provided, further attention to the symptoms is unhelpful—symptoms amplify with attention, and they diminish when attention is drawn away from them (and toward something else) [19,51,53].

As functional neurological symptoms are anxiety provoking for the child and her parents (as well as teachers and other caregivers), attention to symptoms is common. Shifting focus of attention away from symptoms to somewhere else is a key intervention across all components of the biopsychosocial treatment intervention. Focus of attention is worked on in individual psychotherapy with the child, in physiotherapy, on the ward, in the family intervention, and at school.

In individual therapy for example, the child practices the skill of shifting her attention away *from* the symptoms—when she notices her symptoms or notices the warning signs of a functional episode (such as a functional seizure) [20]—*to* the task of implementing a regulation strategy or her functional seizures plan. Similarly, parents are encouraged to shift attention away from symptoms to functional goals (e.g., using regulation strategies, increasing mobility, or returning to school). Some parents need to be coached to allow the child space to practice mobilizing or implementing strategies on her own, and to not provide her with physical assistance or comfort (e.g., stroking, holding, lifting). Psychoeducation about focus of attention and attention to symptoms is also undertaken with the child’s teachers and other school staff. Finally, in the physiotherapy component the physiotherapist is always working to shift attention away from the symptoms or the problem area [86,87].

The following vignettes demonstrate several differing ways that the focus of attention intervention can be integrated within treatment.

Vignette: Becca (carers stepping back to allow development of regulation strategies)

Becca, a 16-year-old girl, suffered from functional seizures (including drop attacks). These were occurring 20–30 times daily. Becca’s parents would provide physical comfort and support: they cradled and stroked her head, lifted her up onto a chair or bed, and asked how she was feeling. In hospital, a new response to functional seizures was negotiated and was practiced by Becca, her parents, and the nursing staff. The nursing staff were to check that Becca was not injured (e.g., had not hit her head) and that she was safe on the floor. Becca was to practice implementation of the regulation strategies she was learning, and she was to get up from the floor when she was ready (had regained control). Over a two-week period, Becca’s functional seizures (and particularly her drop attacks) reduced to approximately 4–5 daily. She and her parents, as well as school staff, were then able to implement the new approach to managing the functional seizures. Becca continued to build her regulation skills with the support of an outpatient therapist.

Vignette: Sarah (use of metaphor to shift attention)

Sarah, a 19-year-old young woman with complex pain and functional weakness in her legs, was struggling to manage escalations of pain. Managing pain was important because the escalations of pain were accompanied by a wobbly gait and difficulties in mobilizing, requiring Sarah to use crutches. Sarah’s therapist used the metaphor of a torch with either a narrow or wide beam to talk about the impact of narrowly focusing on her pain (thereby amplifying her pain) versus expanding her focus to other things (thereby decreasing her pain). Sarah practiced expanding and contracting her focus of attention and became skilled at using this pain management strategy.

Vignette: Finn (parents reduce their attention to symptoms)

Finn, a 13-year-old boy, developed functional seizures following the death of his father and a stressful time at high school. The seizures were unpredictable. Finn’s mother had noticed however, that the seizures seemed to come on days when Finn woke up with a headache. The family doctor suggested that paracetamol would potentially be helpful. Thereafter, each morning on waking Finn, his mother would check his headache in case he needed medication. This wake-up procedure—carried out with good intentions—subsequently became an important *maintenance* factor (daily attention to, and reinforcement of, Finn’s headache) that gave birth to negative expectations of new seizures later the same day. When the wake-up procedure was stopped, Finn’s functional seizures decreased in frequency and finally resolved.

Vignette: Elsa (reducing positive reinforcement)

Elsa, a nine-year-old girl living with both her parents and her little sister, developed acute paralysis in her legs in the aftermath of a serious car accident with her grandparents. Elsa’s parents felt guilty about the accident, and to make up for this, Elsa’s mother made everything nice and cozy at home: fresh baked goods, candlelight, and an endless variety of board games. On getting home from work each day, Elsa’s father was very attentive to her, asking lots of questions about her symptoms. Elsa basked in the attention and told the therapist, “It’s like summer holiday all year around!”.

At first, Elsa’s mother reacted with anger when the therapist explained the role of parental attention and the manner in which this attention reinforced symptoms. However, as the therapeutic relationship grew stronger, Elsa’s parents became “ready” to accept the therapist’s advice. The parents returned to a more everyday-like situation at home. On return home, Elsa’s father found other interests that he and Elsa could discuss together.

Vignette: Tom (carers reduce their attention to symptoms)

Tom, a 12-year-old boy, developed gait difficulties—an uncoordinated gait—when he failed the try out for the elite soccer team in his new school, following a long-distance family relocation. Tom’s symptoms fluctuated. Sometimes he could not stand or walk but could crawl on the floor. At other times—especially when he was alone in his room—the parents noticed that Tom’s legs were functioning. Tom’s physiotherapist had supported the use of a wheelchair, thereby minimizing Tom’s opportunities to mobilize. The paediatrician asked that his parents keep a symptom diary. Several times a day the parents assessed Tom’s symptoms by asking him questions, thereby focusing attention on his symptoms. When the hospital team began to work with Tom, the team asked the parents to stop using the diary, and the physiotherapist in the team began the process of removing the wheelchair.

## 6. Cognitive Behavioural Therapy Interventions

A multitude of treatment interventions targeting cognitions and their associated behaviours sit under the broad umbrella of CBT. These interventions have evolved across three distinct waves, with each wave reflecting “dominant assumptions, methods, and goals” (p. 640) [88,89,90] (see Table 6).

Since many clinicians working in mental health are trained in CBT, they may want to draw on this skill set when working with children with FND. The choice of a CBT intervention—as a component of the broader biopsychosocial treatment intervention—will depend on the biopsychosocial formulation, the skill set of the clinician, and the needs and capacities of the child. A subset of children—for example, children with poor verbal and cognitive skills, or children with developmental delay—may struggle to use the cognitive component of CBT skills. In such scenarios the behavioural component of the intervention may be prioritized, or other adjustments may need to be made.

For clinicians working from a stress-system approach [19,20], children presenting with FND will be first taught to notice—and will practice noticing—the felt sense of the body (see previous section). After that, neurophysiological (bottom-up) regulation strategies will be taught, followed by cognitive (top-down) strategies. The rationale for this sequence is that cognitive strategies are easier to implement when arousal is settled, and when the prefrontal cortex function is no longer disrupted by the physiology of stress [93]. From the lens of CBT, noticing the felt sense of the body reflects a bottom-up mindfulness intervention that has been adapted from Eastern healing traditions [82]. Neurophysiological (bottom-up) regulation strategies—which also build on Eastern healing traditions [82,83,84,85], can be added to the CBT clinician’s toolkit of emotion-regulation strategies (see Wave 3 in Table 6). And cognitive (top-down strategies) are the same as the strategies used in classic and acceptance CBT (see Table 6).

For clinicians using retraining and control therapy (ReACT), a type of CBT, children with FND are instructed to implement a set of behaviours and thoughts (competing responses) as soon as they feel that an episode is about to occur (see Section 6.2 below). From the lens of CBT, this intervention involves a behavioural component (see Wave 1 in Table 6) coupled with a mindfulness component (see Wave 3 in Table 6).

Interestingly, while different theoretical constructs and therapeutic traditions inform the stress-system approach and ReACT, there are significant points of overlap. In both approaches, as soon as the child notices that her body is signalling a functional symptom episode—for example, a functional seizure or a tic event—the child is asked to shift focus of attention away from the symptoms to her treatment plan and to implement “things to do and to think” that help break the symptoms pattern, that promote regulation, and that function to enhance the child’s sense of mastery. 

Detailed below are some examples of CBT interventions used by our teams—and our colleagues—to target illness-promoting cognitive processes. Not all possible interventions are discussed, and for a more detailed explanation, see Kozlowska and colleagues (2020) [19], Fobian and colleagues (2020) [67], and Savage and colleagues (2022) [20].

### 6.1. Using Sequencing to Identify Illness-Promoting Cognitions

Collaboratively sequencing the child’s thoughts and feelings can be a useful technique that helps the child make a connection between thoughts and feelings, expectations, body symptoms of activation, and unhelpful patterns of behaviour, all of which play an important role in triggering or maintaining the FND symptoms. Visual depiction of the sequence on a piece of paper can increase the utility of the intervention and can be used across sessions.

Other names for this kind of sequencing include *functional analysis* in the CBT tradition and *chain analysis* in dialectical behaviour therapy [94,95]. Sequencing can also be used to track body sensations (the felt sense of the body; see section above) and to identify family patterns of behaviour that contribute to symptom maintenance [96,97].

The process of working with the child to develop a detailed sequence can help the child build skills in self-awareness, mindfulness, and mastery. The flowchart below illustrates a visual depiction of a sequence. It illustrates several unhelpful psychological processes: negative/catastrophic thinking, overgeneralization, and low sense of control.

Vignette: Ruth (identifying a sequence)

Ruth was a 14-year-old girl with recent-onset functional tics, characterized by noticeable motor movements and loud vocalizations. Ruth reported that she had no control over her tics, and she often noted that they “come out of nowhere”. Through sequencing (illustrated in Figure 8 below), Ruth was able to identify both physiological (felt sense of the body) and cognitive warning signs in the build up to a “tic attack”.

Ruth and her therapist used the visual representation of the sequence to develop an intervention (illustrated in Figure 9). First, Ruth practiced noticing and labelling her body sensations (warning signs), thoughts, and feelings in real time, as they occurred prior to a tic attack. Second, she began to practice her calming strategies as soon as she noticed the warning signs, and she would allow the attack to pass (whether the body activation eventuated into a tic attack or not). The calming strategies included diaphragmatic breathing, progressive muscle relaxation, and sensory stimulation, including chewing gum, knitting, spinning fidget toys with her fingers, and placing a weighted blanket on her lap. As part of this process, Ruth was also able to practice challenging unhelpful thinking patterns (catastrophizing and overgeneralization). For instance, she was able to replace her thought of “Oh god, what’s that” with “I can feel my warning signs”, and her thought “I have no control” to “I know what to do, I will practice my strategies, and this will pass.”

### 6.2. Habit Reversal

A CBT intervention developed by Aaron Fobian and colleagues for the treatment of functional seizures—termed retraining and control therapy (ReACT)—draws on habit reversal and mindfulness to develop competing responses to functional seizure symptoms and catastrophic thoughts and symptom expectations [67,69,70,71]. When children sense that a functional episode is about to occur, they are asked to attend to their immediate experience (e.g., what they see and hear and their physical sensations) and then to remain aware and conscious of their current experience while engaging in their competing responses to prevent or stop the symptoms of functional seizures. Studies in progress suggest that retraining and control therapy also results in improvements in mixed FND symptoms and in other comorbid functional somatic symptoms (publications in progress).

Vignette: Mariella (using habit reversal to manage functional seizures)

Mariella, a 17-year-old adolescent, developed chest pain (see Box 1, [19,20,98,99,100,101,102]) and leg tremor, following a series of viral infections. Some weeks later she began to experience full-body tremors, sometimes accompanied by memory loss. A diagnosis of functional seizures (a subtype of FND) was given. Mariella found that loud noises, crowds, and making decisions all triggered the episodes, and she began to avoid situations that presented these triggers. With time, the functional seizures worsened, and Mariella was unable to attend school. Daily, she experienced catastrophic-symptom expectations, “This is going to hurt me! I can’t control this! I am going to have an episode every time I am in a crowd!”

At the first retraining and control therapy session, a plan was developed with Mariella to retrain her FND symptoms. Competing thoughts—“I’m safe. These episodes do not hurt me. I have a good plan to manage my symptoms.”—were developed to counter her catastrophic symptom expectations. Mariella was given opposing responses to each of her symptoms (e.g., moving arms and legs in and out to counter tremors, focusing on items around her and describing them aloud to counter perceived loss of consciousness, speaking or singing to counter vocalizations), and she was instructed to begin these as soon as she felt an episode was about to begin and to continue doing it after the symptoms started, with the aim of preventing or stopping the episode. Mariella’s parents were instructed to ensure her safety during the episodes but to allow her to manage the episode independently. A school reintegration plan was made and initiated.

At the second session, Mariella reported the ability to shorten episodes to about eight minutes. The frequency of her functional seizures had reduced. At the third session, she indicated that she was able to prevent episodes before they started, but she reported the occurrence of a new symptom, head jerking. Rolling her head was added as an additional opposing response for this symptom. Motivators were developed as rewards for meeting the goals that she developed for treatment (e.g., returning to school full time). Over sessions 3–5, Mariella’s symptoms continued to decrease. She returned to school full time. During session 6–9 the cognitive behavioural techniques used to manage the functional symptoms were generalized to other situations, helping her to develop skills for managing general life stressors. At one year follow up, Mariella was still doing well.

Box 1Hyperventilation-related chest painChest pain is a common functional symptom caused by hyperventilation-related constriction of the coronary arteries [98,99,100]. Hyperventilation reflects activation of the motor-respiratory system [19,20]. It is common in children and adolescents with functional seizures [101,102]. It results in hypocapnia—low levels of arterial carbon dioxide and a respiratory alkalosis—that is responsible for a broad range of functional somatic symptoms, including hyperventilation-related chest pain.

### 6.3. Exposure Interventions to Manage Avoidance of Difficult Thoughts, Feelings, and Situations

Avoidance processes are common in children with FND (see results of the current study in Table 5). For some children, the process of avoiding difficult thoughts (cognitive avoidance) or difficult feelings (feeling avoidance) functions to maintain their symptoms. In this context, learning to tolerate and sit with these thoughts and feelings, although difficult, can be helpful. Alternately, many children with FND present with an established pattern of behavioural avoidance—for example, withdrawing from situations or places that might trigger their functional symptoms. “This avoidance is unhelpful because it establishes a vicious cycle: the young person does less and less; the young person loses her sense of mastery and becomes more and more deconditioned; and the functional (symptoms) increase in frequency” (p. 117) [20]. In this context, learning to successfully manage situations or places that have been avoided is likewise an important element of the intervention.

The intervention of sitting with difficult thoughts and feelings is one that borrows from Eastern meditative traditions (and acceptance and commitment therapy in Western psychology) of mindfully accepting and allowing uncomfortable or unpleasant thoughts and feelings to come and go without struggling against them. The intervention also uses some of the principles of habituation from behavioural therapy; that is, by sitting with an unpleasant thought or feeling, body arousal will decrease in unpleasantness with time. It can be useful to incorporate the analogy of “surfing the wave” here to provide a visualization for the child to imagine while practicing this challenging skill.

Vignette: Safia (using mindfulness to sit with difficult feelings and thoughts)

Safia, a 15-year-old girl, started experiencing episodes of sudden twitching and shaking upon return home from school camp, where she had contracted the flu. Safia had great difficulty identifying any warning signs to her functional episodes, often reporting they “came out of nowhere”. She also struggled to identify any of her emotions or thoughts. A number of interventions were implemented to help Safia. A mindful body-scan task was introduced to support Safia to connect with the felt sense of her body and the sensations that she could feel arising and ebbing in her body. A feelings wheel helped Safia identify different feelings. Using coloured pencils, Safia then represented these feelings on a body map. In so doing, she made a link between her feelings and the physiological sense of her body. Finally, a mindfulness task helped Safia to notice her thoughts and feelings as they came into consciousness, to observe them without judgment, and to let them go. With repeated practice, Safia eventually was able build her capacity to notice when her functional twitching and shaking episodes were about to happen. She noticed her triggers included feelings of anger or a negative thought that lingered in her mind, and that her physiological warning signs included an increase in breathing rate, sweaty palms, and a sense of dizziness.

The use of an exposure hierarchy intervention, which gradually exposes the child to feared situations or activities in a safe therapeutic space, works to break the avoidance cycle and support the pathway back to regular functioning. Importantly, exposure treatment should be coupled with one or more coping strategies. This could include a neurophysiological (bottom-up) regulation strategy (e.g., paced breathing or progressive muscle relaxation), a top-down strategy such as thought challenging (e.g., practicing a positive statement), and a focus-of-attention strategy, (e.g., switching attention to a secondary task such as a fidget toy or listening to a piece of music). Alternatively, exposure intervention could include a body-oriented mindfulness approach with a focus on sitting with, tracking, and tolerating difficult sensations (an approach used in many body-oriented therapies and in third-wave CBT, including acceptance and commitment therapy). The broader biopsychosocial formulation ensures that the exposure intervention is combined with other appropriate interventions: addressing bullying if bullying is driving the avoidance, addressing learning difficulties if these are driving the avoidance, and addressing other safety issues if these are driving the avoidance.

Vignette: Corey (using exposure to widen the social context)

Corey was an 11-year-old boy with severe functional seizures that manifest as violent, full-body jerking that sometimes caused him to catapult out of his bed. Corey had been bed bound for many months. Attempts to widen his social context beyond his bed—to enable discharge from the hospital—were met with anxiety that triggered a functional seizure. The graded exposure hierarchy began with trips to the hospital café, where Corey and his father drank milkshakes and played rounds of uno. Despite Corey repeatedly stating “it won’t work” and “I’ll stuff it up”, the exposure hierarchy was extended to cafés outside the hospital. Eventually, it was taken home by the family and integrated into a very gradual transition plan back to school (starting with driving past the schoolground with his parents).

## 7. Trauma-Processing Interventions

For a subset of children, intrusive memories or thoughts related to unresolved loss or an experience of trauma can exacerbate and maintain their functional symptoms (see Table 5). For this group, trauma-specific interventions such as eye movement desensitization and reprocessing (EMDR) [103,104], trauma-focused CBT [104], radical exposure tapping (RET) [105], and accelerated resolution therapy (ART) [106] may be helpful.

Vignette: Morgan (use of eye movement desensitization and reprocessing)

Morgan was a 14-year-old female with a history of complex PTSD and anxiety stemming from being exposed to chronic and violent domestic abuse perpetrated by her biological mother’s boyfriend while Morgan was 1–4 years of age. Morgan went to live with her grandmother at age 5. Through the loving and stable environment that her grandmother provided, Morgan thrived. When Morgan was 13, she was at a local bakery with her friends when a song came on the radio that the perpetrator of the domestic violence had often played in the home. Morgan became hypervigilant. She responded with panic attacks to any song on the radio that sounded similar. Two weeks after the event, Morgan started experiencing functional seizures, characterized by prolonged periods of unresponsiveness. As part of her treatment, Morgan engaged in EMDR therapy, alongside other mind–body interventions—to assist in down regulating her stress system (so that the functional seizures could resolve).

Vignette: Deepa (use of radical exposure tapping)

Deepa’s first FND episode (weakness in the legs and ataxic gait) was triggered at 14 years of age by the shock of death. First, her rabbit died, and then a few weeks later, her grandfather. Her second FND episode (leg weakness and functional seizures) at 15 years of age was triggered by her father’s near-death experience on the operating table and ten-day admission to intensive care. Her third FND episode (jaw dystonia and functional seizures) was triggered after the family had visited her grandfather’s grave. Her forth FND episode (jaw dystonia) was triggered when she heard that her father needed another hospital admission for a minor procedure. Deepa and her therapist used RET first with the memories of her rabbit, then her grandfather alive, then her grandfather in his coffin, a visual image of the current grave site, and then memories of Deepa’s father, his stay in intensive care, and his coming home, to process Deepa’s loss and trauma memories.

## 8. Hypnosis

As noted previously, top-down regulation strategies are those that involve use of the mind—involving of imagery and metaphor, as well as self-directed shifting of attention, thoughts, or feelings—to bring about change to brain and body states. Many top-down interventions for FND—and other functional somatic symptoms—can be considered as incorporating forms of hypnosis, “a unique form of top-down regulation [i.e., information processing] in which verbal suggestions are capable of eliciting pronounced changes” in neurophysiology, perception, and behaviour (p. 59) [107].

According to Michael Yapko, hypnosis involves “a focused experience of attentional absorption”—sometimes referred to as *trance*—that invites the child to pay “greater attention to the essential skills of using words and gestures in particular ways to achieve specific therapeutic outcomes” (p. 8) [108]. In daily clinical practice, health-promoting suggestions, imagination, creativity, and metaphor can be integrated into the natural conversation with the child and family [109]. Clinicians can use suggestions—verbal or nonverbal—to invite the child to experience herself or the world in a new way, thereby leading to changes in the child’s physiology, sensations, perception, emotions, thoughts, and behaviours [109]. Alternatively, suggestions can be embedded in the ritual of the treatment intervention, including formal hypnosis sessions. Calming the body through the practice of hypnosis and other mind–body regulation strategies can serve to promote feelings of agency, control, and mastery, and to promote the healing process and the child’s journey to health and wellbeing. For more detail about the use of hypnosis in FND and other functional somatic symptoms, see Helgeland and colleagues [109].

Vignette: Sammy (using hypnosis as calming strategy)

Sammy, a 13-year-old girl who was very reactive in social situations, would become very angry and distressed when her friends did not respond in a way that she perceived to be supportive. This would then activate her stress system and escalate her functional symptoms. Sammy would then blame her friends for making her sick. Sammy’s therapist used a brief hypnosis session in which Sammy practiced being the “eye of the storm” with emotions and reactions swirling around her. Sammy then practiced bringing up imagery that helped to calm the storm. Sammy also found a smooth pebble she liked holding. She would keep this in her pocket at school and touch it as a way to help her remember to visualize water swirling around her—just like difficult emotions would do. Sammy was subsequently able to access this imagery and the associated feelings of calm when she was triggered by her social setting. She used this interval of calming imagery to settle herself before she acted.

Vignette: Malin (using hypnosis to imagine the past and visualize a new future)

Malin, a 16-year-old girl with paralysis in both legs, had lost all hope for recovering and did not believe that a therapist could be of any help. “You see, my body doesn’t remember how it feels to walk, dance, feel energized, or just be happy.” The therapist used hypnosis to help Malin experience how she could use the power of her imagination in the present to remember the past and to sow a seed in her mind about a positive illness outcome. Guided by the therapist, Malin used her imagination to recall in detail a wonderful situation from the past in which her legs were doing their job all by themselves while she just was happy and had a nice feeling of energy and of control. Malin was amazed by how her mind was able to evoke all the pleasant sensations, emotions, and memories in her body. “I wonder if your mind’s power also can reach out to the future?”—the therapist wondered aloud. Malin was curious, and again, guided by the therapist, she used her imagination to create a wonderful future situation where she was at a marvellous party with her friends, dancing, laughing, and feeling happy—while her strong legs were doing their job all by themselves. This contributed to a shift in Malin’s expectation for the future symptom course and made her more open to therapeutic change and eventual recovery from her FND.

## 9. Family Interventions That Scaffold the Work with the Child

Given that many of the illness-promoting psychological processes are shared by, or related to, interactions within a child’s family system, inclusion of the family is integral to the biopsychosocial treatment intervention. Family interventions will typically occur in parallel with the child’s individual therapy work. In the initial stage of treatment, family interventions tend to be more directive, which can shift to a more reflective approach once some initial gains have been achieved. Common initial (shorter-term) goals with families include containing parental anxiety, decreasing or eliminating parental attention to symptoms, direct coaching around management of FND symptoms, addressing iatrogenic trauma, and returning to normal function. Longer-term goals are wide ranging, and in some clinical scenarios, a family-based approach will be the cornerstone of the holistic (biopsychosocial) treatment plan.

In short, when individual therapy is offered to the child, it is imperative that the parents and broader family understand and respect the need for such an approach and are able to allow and support the child’s efforts to change within the frame of the family system. Sometimes clinicians can observe that individual work with a child fails because the parents do not understand, agree with, or support the rationale for such an approach. In this scenario, the family system does not allow or is not open for therapeutic changes in the child.

For more information see Chapter 12, “Working with the Family”, in Savage and colleagues (2022) [20].

## 10. School Interventions That Scaffold the Work with the Child

As many of the illness-promoting psychological processes play out in the school setting, school interventions that scaffold the work with the child are integral to the treatment program. The key challenge for schools is to provide support—in accordance with the individual child’s need—without focusing on the functional symptoms. The school may also need to address school-related stress: unrealistic expectations for academic achievement, the stress associated with examinations and the academic process, learning difficulties, and difficulties in peer relationships. For more information about school interventions, see Chapter 13, “Working with the School”, in Savage and colleagues (2022) [20] and Online Supplement 16.3, “Working with the School”, in Kozlowska and colleagues (2020) [19].

## 11. Discussion

The current study documents the illness-promoting psychological processes that were part of the clinical presentation of children admitted for a biopsychosocial multidisciplinary inpatient intervention—the Mind–Body Program—for treatment of FND. The study also documents subjective distress (total DASS score), ACEs reported by the children (family interview and ELSQ), and level of function on admission (GAF score). The most common illness-promoting psychological processes (assessed only in children with FND) clustered into the following domains: attention to symptoms; cognitive processes pertaining to the symptoms per se (e.g., negative/catastrophic-symptom expectations and low sense of control); cognitive processes pertaining to life challenges in general (e.g., self-critical rumination); feeling-related processes (feeling worried or sad); avoidance processes (pushing difficult thoughts and feelings out of mind and avoidance of activities); inability to track body state; and intrusive thoughts/feelings/memories reflecting unresolved loss, unresolved trauma, and unresolved negative experiences. Compared to controls, children with FND reported more ACEs across the lifespan (higher ELSQ score) and more subjective distress (higher total DASS score). They showed significant impairment in global function—lower GAF scores—consistent with their disabling FND symptoms. In the discussion that follows, our reflections pertain to our findings from our particular cohort of patients, who were sufficiently ill to require an inpatient intervention, and on the authors’ clinical experience working with paediatric patients with FND in an inpatient rehabilitation setting.

The most common ACEs that emerged during family interview included bullying or rejection by peers, physical illness experienced by the child, and family conflict. Both the actual experience of social rejection and subsequent rejection-related thoughts and feelings function to activate neural networks that mediate the subjective experience of pain (>2/3 of the current cohort) [110,111,112] and stress-system activation (autonomic system component) [25], both of which are known to modulate aberrant neural network function in paediatric FND [39,113]. On the ELSQ, the reported rates of physical and sexual abuse were also low, but one-fourth of children reported emotional abuse that they had experienced across family, school, and other social settings, and one fifth reported that they had been exposed to domestic violence. In addition, more than a quarter reported prematurity or birth complications and more than a quarter had experienced surgery or repeated hospitalizations. And one fifth reported that they had experienced a natural disaster first-hand, reflecting the cumulative natural disasters—drought, floods, storm, and bushfires—that have afflicted the state of New South Wales in recent years [114,115].

All of these ACEs implicate stress-system activation—and priming—from early in life, via a complex web of processes (see Chung et al. for a comprehensive discussion pertaining to FND) [38,41,116]. These processes enable ACEs to be biologically embedded in body and brain, thereby increasing risk for FND and for other functional, medical, and mental health disorders [38,117,118]. Taken together, these data suggest that ACEs are broad ranging and that activation and priming of the stress system—leading to subsequent vulnerability to FND—take place across the lifespan [37,38,41,43,119]. The trigger event that initially activates the FND may be relatively minor. These data also highlight that illness-promoting psychological processes arise and are embedded in the child’s lived experience.

The illness-promoting psychological processes identified in the cohort were consistent with the life challenges and predicaments faced by this group of children (see paragraph above, and Table 2 and Table 4). A subset of psychological processes pertained to the challenges that the children faced in terms of managing their FND symptoms: attention to symptoms, negative/catastrophic-symptom expectations, and low sense of control in relation to the symptoms. Another subset of psychological processes pertained to ongoing life challenges. For example, the children’s high and unrealistic expectations of themselves and sometimes also the expectations held by others were expressed as self-critical rumination, catastrophic thinking, and perfectionistic thinking. Other illness-promoting processes—such as feeling worried or sad, or intrusive thoughts/feelings/memories—were tied in with the multiple ACEs that the children had thus far experienced and had needed to manage. And avoidance processes, such as pushing difficult thoughts and feelings out of mind and avoidance of activities (including school), were linked to the children’s attempts to avoid distressing thoughts, feelings, and activities—ones that the children did not have the capacity, skills, and support to manage.

While only a small percentage (13.2%; *n* = 10) of children met diagnostic criteria for PTSD, the alternating psychological processes of being overwhelmed by thoughts, feelings, memories associated with negative events and trying to push them out of mind were also found in the broader sample (see feeling-related processes and avoidance processes, Table 5). In a different cohort of children with FND, this same psychological pattern was identified in a study of attachment (see Introduction) using linguistic analysis of attachment transcripts [56]. These findings suggest that identifying and treating illness-promoting psychological processes—whether the child meets criteria for a comorbid mental health diagnosis of not—is a useful element of the therapeutic process and the path toward health and wellbeing [78]. The long-term goals of addressing illness-promoting psychological processes are to increase capacity to manage stress, improve resilience, and build a skill base that empowers the child to manage future life challenges in a more effective way.

Of particular importance is that each and every one of the identified illness-promoting psychological processes is a potential target of therapeutic intervention (see Section 4 above). Moreover, because subjective distress—which is driven and amplified by illness-promoting psychological processes—contributes to stress-system/neural network dysregulation that maintains FND symptoms [38,39], the psychological component of the treatment intervention is central to the therapeutic process. Put simply, addressing illness-promoting psychological processes therapeutically—in work with child, family, and school—functions to address mind–body processes that trigger and maintain FND symptoms. In our experience, however, the necessary information about psychological processes tends to emerge slowly during the therapeutic process, and it is unusual to be able to access information about inner thoughts and feelings via brief evaluation processes (including questionnaires) [65]. For this very reason we utilized a unique prospective methodology which allowed us to identify illness-promoting psychological processes that emerged within the therapeutic space—including the child’s work with a therapist in daily psychotherapy sessions—over a two-week period within an inpatient setting (Mind–Body Program admission).

Finally, as noted throughout the article, in paediatric practice the biopsychosocial formulation—based on a comprehensive assessment—needs to inform the child’s overall treatment plan. Interventions that address illness-promoting psychological processes are just one component of the biopsychosocial intervention. While much of this work may be done with the child, scaffolding from parents and school staff is always of the upmost importance. Likewise, addressing family, school, peer, medical, and comorbid mental health issues that contribute to the clinical presentation—as identified by the biopsychosocial formulation—is essential.

The current study has a number of limitations. First, the list of illness-promoting psychological processes presented in Table 5 was developed for clinical purposes, to be used in the Mind–Body Program. The processes—which are overlapping—were not subjected to a formal factor analysis that would potentially have reduced them to a smaller number of factors, resulting in a more robust and shorter list. In this context, the study is exploratory: it begins the work of identifying the psychological processes that typify this group of patients. Second, the illness-promoting psychological processes reported in this study were identified by the mind–body team in collaboration with patients during their two-week inpatient admissions. The inpatient context provides many opportunities for illness-promoting psychological processes to emerge and to be identified. These opportunities include multiple interactions with the treating team (daily ward rounds, individual therapy sessions, physiotherapy sessions), interactions with nurses, interactions with school staff, and so on. All of these interactions create a safe therapeutic space [72,86]—a context that supports the process of opening up about inner thoughts and feelings. It is unlikely that a simple enquiry—or the use of a questionnaire at the onset of the admission process—would have elicited such a rich dataset. For many children, becoming aware of illness-promoting processes—and addressing them—was part of the treatment intervention, both during and after the mind–body admission. Third, the children who participated in the current study were sufficiently ill to justify admission for treatment to an inpatient setting. In addition to substantial functional impairment in the context of FND, many experienced comorbid pain, comorbid functional somatic symptoms (not FND), and comorbid psychiatric disorders. In this context the findings of the current study may not be reflective of children who are less ill or who present with transient FND symptoms.

Notwithstanding, the findings of this study are important. Taken together, the study suggests that our cohort of children with FND had experienced cumulative stress across the lifespan—physical, relational (family or peer related), academic, or ecological. This cumulative stress was associated with dysregulation of the stress system and neural networks [38,39], high levels of subjective distress, and activation of psychological processes that typically arise when children’s coping capacities—body and mind—have been dysregulated or overwhelmed. These illness-promoting psychological processes play an important role in the development and maintenance of FND in children and should be a target for intervention.

## 12. Conclusions

In conclusion, the current study highlights that illness-promoting psychological processes are common in children disabled by FND and that they emerge from, and are embedded in, the challenges of daily living faced by the child and child’s family, and in the school and socio-ecological context (see Figure 3). Illness-promoting psychological processes contribute to the child’s ongoing sense of subjective distress and, if not addressed, can maintain the illness process. The ACEs experienced by children with FND are wide ranging and cumulative in nature. They are hypothesized to activate and prime the stress system—including brain stress systems that underpin salience detection, arousal, pain, and emotional states—and to reflect the biological embedding of experience that makes the child’s brain, body, and mind more likely to respond to future stress with an amplified response [37,38,41,119]. Illness-promoting psychological processes are part the psychological manifestation of this amplified defensive response. Addressing illness-promoting psychological processes therapeutically—using a range of clinical interventions—facilitates positive expectations, enables the child to recognize and respond to body signals of stress-system activation, provides skills for managing focus of attention, and builds the necessary skill base for managing other illness-promoting psychological processes and promoting resilience in the face of future life challenges.

## Figures and Tables

**Figure 1 children-10-01724-f001:**
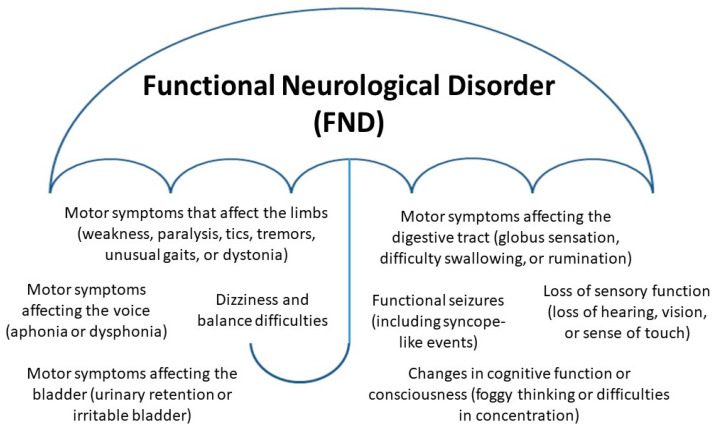
**Symptoms falling under the umbrella of functional neurological disorder.** Visual representation showing the broad range of symptoms that sit under the umbrella of functional neurological disorder. Reproduced with permission © Kasia Kozlowska 2019 [20].

**Figure 2 children-10-01724-f002:**
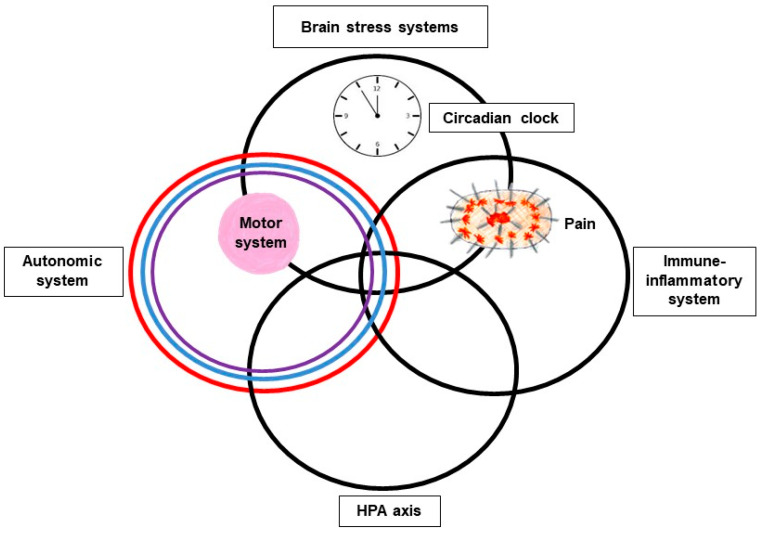
**The stress-system model for functional somatic symptoms: Circles metaphor.** The overlap between the different components of the stress system—the HPA axis, autonomic nervous system, immune–inflammatory system, and brain stress systems—is presented by the overlap between the circles. The brain stress systems refer to the areas of the brain that are involved with salience detection, arousal, pain, and emotional states. The circadian clock is placed within the top circle because the master clock is found in the hypothalamus, a small region located in the base of the brain. The motor system, which includes central and peripheral components, is represented by the pink ball. The placement of the pink ball in the overlap between the brain stress systems and autonomic system reflects that activation of these systems can be accompanied by changes in motor function. The pain system, which also includes central and peripheral components, is represented by the spiky oval. The placement of pain in the overlap between the brain stress systems and immune–inflammatory system reflects that activation of these systems maintains chronic pain. HPA = hypothalamic–pituitary–adrenal. Reproduced with permission © Kasia Kozlowska 2013 [36].

**Figure 3 children-10-01724-f003:**
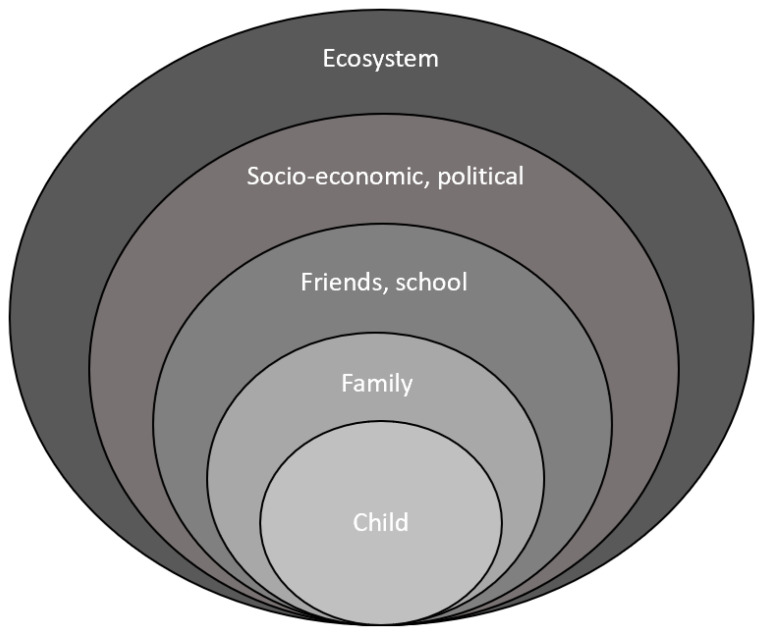
**Biopsychosocial and ecological system levels.** The biological psychological, socio-economic, political, and ecological system levels in which the child is embedded and that effect the child’s health and wellbeing. © Kasia Kozlowska and Olivia Schollar-Root, 2023.

**Figure 4 children-10-01724-f004:**
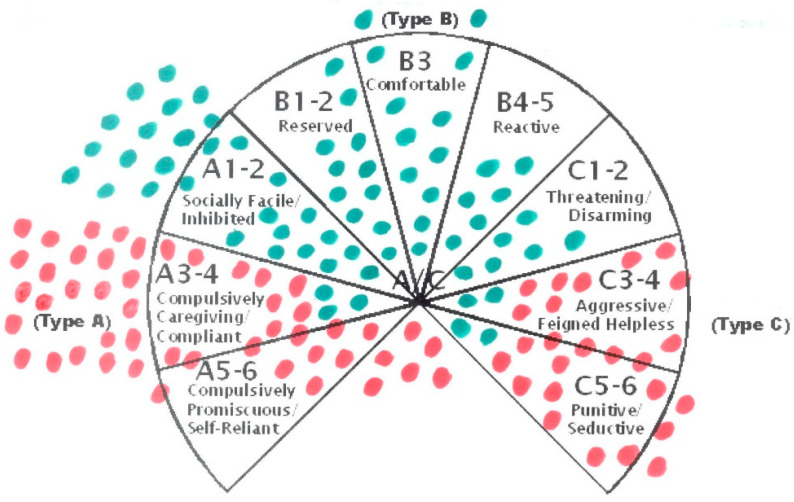
**Patterns of attachment in children with functional neurological disorder.** The healthy controls (green dots) were classified into the normative patterns of attachment (A1-2, B1-5, C1-2), and the children with functional neurological disorder (the red dots) were classified into the at-risk patterns of attachment strategies, the A+ inhibitory cluster (A3-4, A5-6) and the C+ coercive cluster, C3-4, C5-6). The rates of unresolved loss and trauma assessed via linguistic markers were 75% (*n* = 57/76) of children with functional neurological disorder versus 12% (*n* = 9/76) of controls. The dynamic maturational model depicting patterns of attachment is reproduced with permission of Patricia Crittenden, © Patricia M. Crittenden, 2001.

**Figure 5 children-10-01724-f005:**
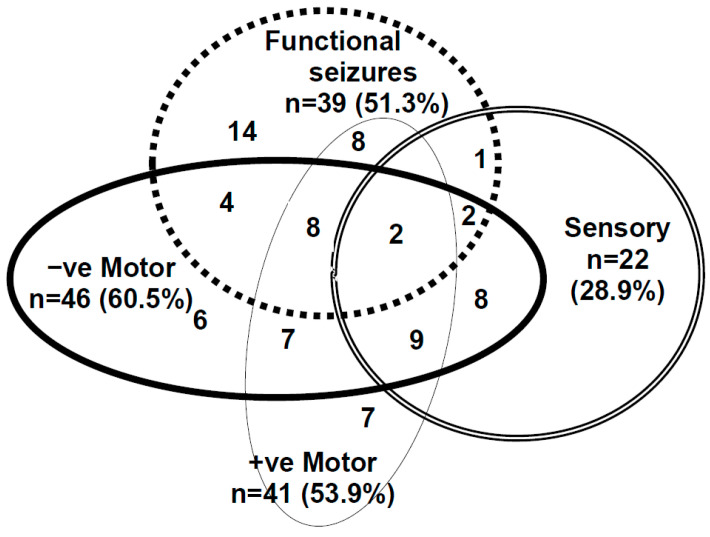
**The different types of functional neurological symptoms experienced by the 76 children in the cohort.** The majority of children presented with multiple functional neurological symptoms (mixed FND, *n* = 49; 64%). Negative motor symptoms included the following: weakness or loss of function in the limbs (*n* = 46; 61%), aphonia (loss of voice; *n* = 2; 3%), and difficulties swallowing (*n* = 3; 4%). Positive motor symptoms included the following: unusual/uncoordinated gaits coupled with difficulties with balance (*n* = 18, 24%), tics (*n* = 16; 21%), tremors (*n* = 18; 24%), dystonia (*n* = 5; 7%), rumination (bringing up food via overactivation of the diaphragm; *n* = 6; 8%), and dysphonia (change in the quality of the voice, e.g., a high-pitched baby voice, *n* = 5; 7%). Sensory symptoms included the following: loss of touch (*n* = 9; 12%), loss of hearing (*n* = 5; 7%), or loss of vision (*n* = 11; 14.5%). Functional seizures presented in a broad variety of ways and included faint-like events (*n* = 39; 51%). Not included in this representation are children who presented with loss of memory (*n* = 6; 7.9%)—a cognitive symptom—as part of their clinical presentation (see Table 2).

**Figure 6 children-10-01724-f006:**
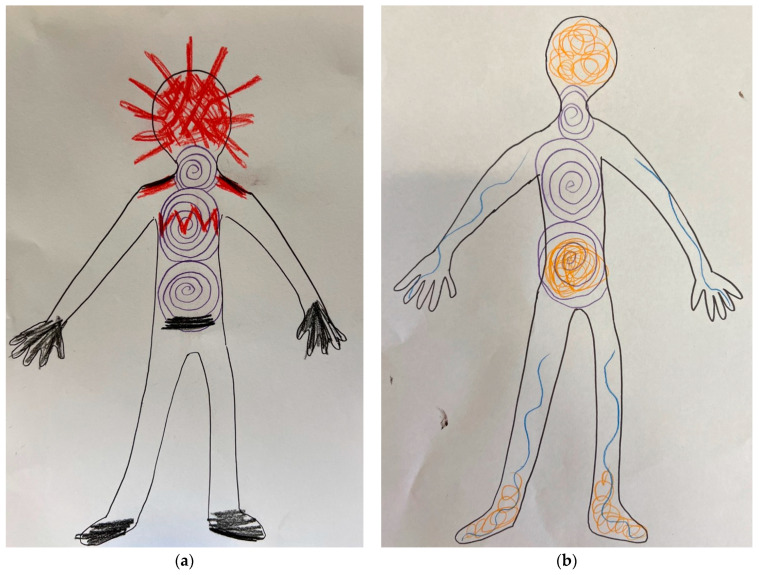
This body map, drawn by an adolescent girl, represents her body in different states of activation. Image (**a**) reflects her “warning signs” of an impending functional seizure or tic event (she suffered from both). Image (**b**) represents her state of greater calm following implementation of her regulation strategies. Reproduced with permission. © Kasia Kozlowska 2022.

**Figure 7 children-10-01724-f007:**
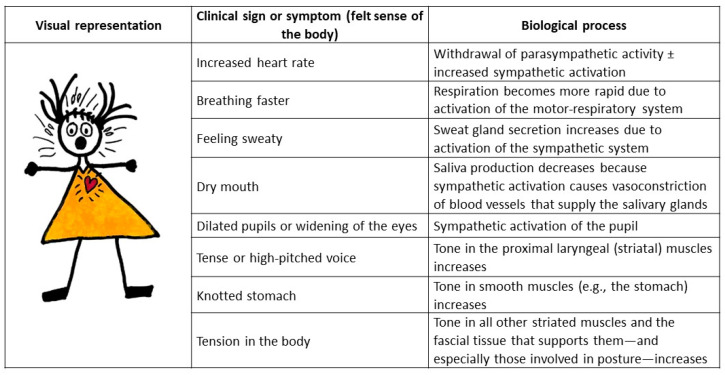
**Common signs and symptoms associated with increased arousal.** Reproduced with permission from Savage and colleagues (2022) [20] © Kasia Kozlowska 2019.

**Figure 8 children-10-01724-f008:**
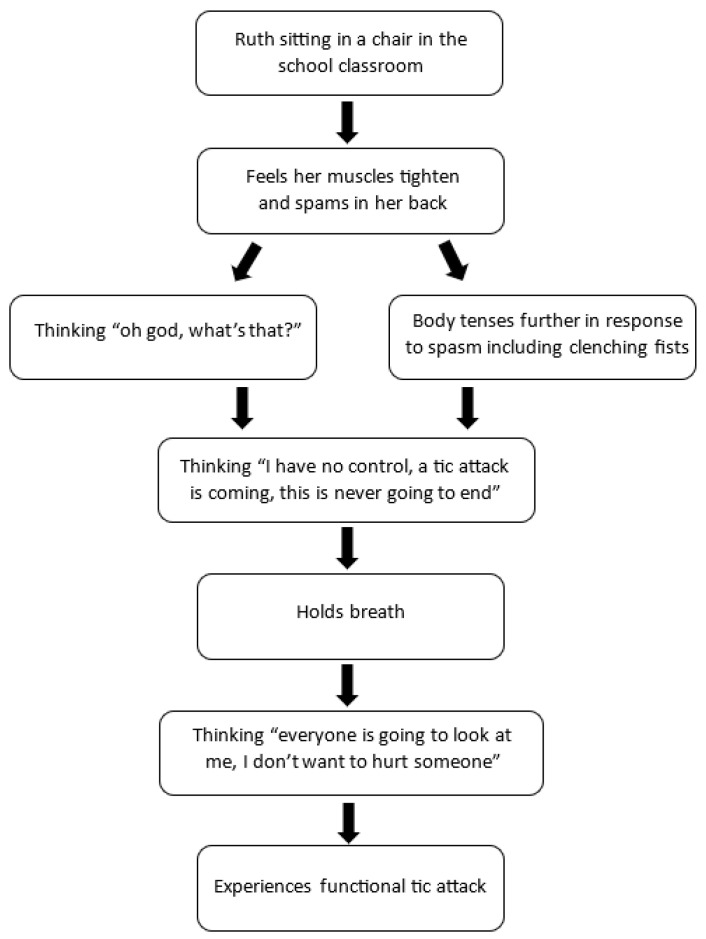
A visual representation of the sequence of events leading up to a tic attack, as identified by Ruth and her therapist.

**Figure 9 children-10-01724-f009:**
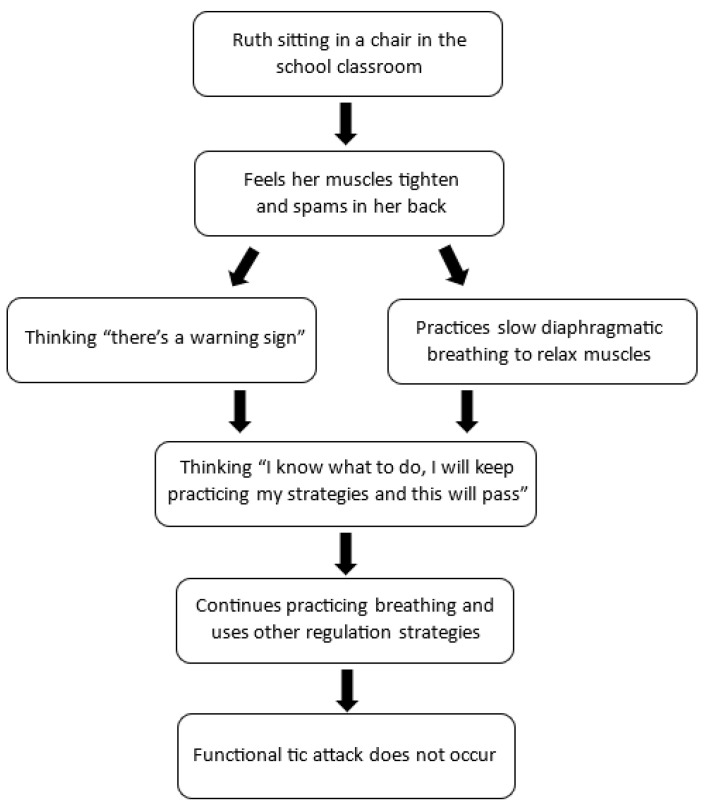
A visual representation of the sequence of events, coupled with Ruth’s implementation of regulation strategies to calm her body system, after she has identified her warning signs of a tic attack.

**Table 1 children-10-01724-t001:** Summary of the Measures Used in the Study.

Measure	Description
RAHC-GAF	The Royal Alexandra Hospital for Children Global Assessment of Function is the DSM-IV-TR Global Assessment of Functioning modified to include functional impairment secondary to physical illness [74]. The scale has 100 points and 10 categories (10 points each). Healthy controls fall into the upper two brackets “superior in all areas” (score 91−100) or “good in all areas” (score 81−90). Lower values (and brackets) mark functional impairment of increasing severity. Patients with physical or psychological impairment fall into the lower brackets (score < 81).
DASS-21	The Depression Anxiety and Stress Scales are a validated measure of perceived distress in paediatric populations [75,76].
ELSQ	The Early Life Stress Questionnaire is a checklist of 19 stress items—and an option for elaboration—based on the Child Abuse and Trauma Scale [77]. Twelve of the items pertain to relational stressors: bullying, physical abuse, sexual abuse, emotional abuse, neglect, parental separation, loss by separation, loss by death, family conflict, severe illness of a family member, domestic violence, and other. Other items pertain to birth complications, life-threatening/severe illness, war trauma, and natural disasters. Participants record if they have or have not experienced the given stressor and the age period during which the stressor has been experienced.

**Table 2 children-10-01724-t002:** Clinical and demographic information about the 76 participants with FND from clinical assessment (Version C).

**Comorbid Medical Conditions**		
Any comorbid medical condition ^a^	30	39.5%
− Asthma/allergies	8	10.5%
− Deficiencies (iron, vitamin D, vitamin B12)	7	9.2%
− Hypermobility	5	6.6%
− Epilepsy	3	3.9%
− Tourette’s syndrome	2	2.6%
− Kidney disease	2	7.6%
− Coeliac disease	2	7.6%
− Miscellaneous other	8	10.1%
**Comorbid Functional Syndromes**		
Any complex (functional) pain	51	67.1%
− Headache	26	34.2%
− Abdomen	14	18.2.0%
− Lower limbs	12	15.8%
− Back/neck	12	15.8%
− Chest	6	7.9%
Any comorbid functional syndrome (excluding pain)	23	30.3.8%
− Postural orthostatic tachycardia syndrome	18	23.7%
− Functional gastrointestinal disorder	8	10.5%
− Irritable bladder	2	2.6%
**Comorbid Nonspecific Somatic Symptoms**		
Any comorbid nonspecific somatic symptom	68	89.5%
− Fatigue	42	55.3%
− Dizziness	41	53.9%
− Nausea	36	47.4%
− Heart palpitations (thumping heart)	29	38.2%
− Breathlessness	22	28.9%
− Sweatiness	18	23.7%
**Comorbid Nonspecific Symptoms Other**		
Sleep (difficulties falling asleep, waking, unrefreshing sleep)	61	80.3%
Concentration difficulties	52	68.4%
**Comorbid Mental Health Disorders And Symptoms**		
Any mental health disorder (DSM-5)	66	86.8%
− Anxiety disorder (generalized, panic disorder, OCD)	61	80.3%
− Depressive disorder	33	43.4%
− PTSD	10	13.2%
− ADHD	5	6.6%
− Autism spectrum disorder	8	10.5%
− Learning difficulties	7	9.2%
− Psychosis	1	1.3%
**Common Adverse Childhood Experiences (ACEs) Reported by the Child and Family (Maltreatment-Related Events Are Denoted by an Asterisk)**		
One or more ACEs (range = 1–12; mean = 5.79; median = 6.00)	76	100%
− Bullying by peers	47	61.8%
− Child physical illness ^b^	41	53.9%
− Family conflict	36	47.4%
− Maternal mental illness	33	43.4%
− Loss via separation from a loved one or a close friend	26	34.2%
− Loss via death of a loved one	25	32.9%
− Paternal mental illness	20	26.3%
− Maternal physical illness	17	22.4%
− Exposure to domestic violence *	16	21.1%
− Paternal physical illness	15	19.7%
− Multiple moves of house	17	22.4%
− Emotional abuse (e.g., rejection/abandonment by a parent, parent emotional reactivity post brain injury) *	11	14.5%
− Physical abuse *	7	9.2%
− Sexual abuse *	7	9.2%
− Neglect *	5	6.6%
− Prolonged custody battle	5	6.6%
− Migration stress	3	3.9%
**Socioeconomic Status of the Family**		
Professional	5	6.6%
White collar	18	23.7%
Blue collar	25	32.9%
Unemployed	2	6.5%
**Family Constellation**		
Biological parents	47	61.8%
Blended family	25	32.9%
Lives with one parent (single mother in these cases)	3	3.9%
Foster care	1	1.3%
**Intelligence Quota Estimated from School Reports (or Formal Testing)**		
Superior range (120+)	20	26.3%
Average range (80−119)	44	57.9%
Borderline range (70−79)	9	11.8%
Delayed (<70)	3	3.9%

^a^ Some children had more than one medical condition. ^b^ Denotes a physical trigger (an illness [*n* = 13; 17.1%] or a fall/injury [*n* = 7; 9.2%]), a medical procedure (*n* = 2; 2.6%), and vaccination (*n* = 4; 5.3%) were the trigger events that immediately preceded the child’s FND illness in 26 (34.2%) of cases. * Denotes maltreatment events. A third of children (*n* = 27; 35.5%) and families reported that the child had experienced some form of maltreatment (physical abuse, sexual abuse, neglect, emotional abuse, or exposure to domestic violence). In half of this subset of cases (*n* = 14; 18.4%), the maltreatment had been formally documented in the Child Protection System. ADHD, attention-deficit/hyperactivity disorder; OCD, obsessive-compulsive disorder; PTSD, posttraumatic stress disorder.

**Table 3 children-10-01724-t003:** Comparisons between FND and healthy-control groups on Global Assessment of Function (GAF), Depression Anxiety and Stress Scales (DASS-21), and Early Life Stress Questionnaire (ELSQ) (Version A).

Measure	FND Group (*n* = 76): ^a^ Mean Value/Total Score (Range)	Healthy-Control Group (*n* = 47): Mean Value/Total Score (Range)	*t*/χ^2^ (*p*)
GAF	32.04 (10–51)	89.66 (75–99)	−42.40 (<0.001)
DASS-21 Total Score	26.82 (2–52)	5.68 0–30	12.18 (<0.001)
ELSQ	3.64 (0–10)	0.51 (0–3)	9.38 (<0.001)

^a^ Four participants with FND were missing from the DASS-21 analysis, and three participants with FND from the ELSQ analysis.

**Table 4 children-10-01724-t004:** Responses of children with FND on the Early Life Stress Questionnaire (Version A).

Item (19 Items)	Number (*n* = 73)	Percentage
Prematurity/birth complications	21	27.6
Adopted	0	0
Major surgery or repeated hospitalization	20	26.3
Life-threatening illness/injury (self)	5	6.6
Sustained bullying of rejection by school mates	41	53.9
Physical abuse	11	14.5
Sexual abuse	6	7.9
Emotional abuse	20	26.3
Extreme poverty	4	5.3
Witness natural disaster first-hand	15	19.7
House destroyed by fire or other means	5	6.6
Witness warfare	0	0
Parents divorced or separated	27	35.5
Separated long time from a parent, brother, or sister	12	15.8
Sustained conflict within family	23	30.3
Death of a parent, brother, or sister	7	9.2
Life-threatening illness (parents, brother, or sister)	10	13.2
Witness domestic violence in family	14	18.4
Witness of experience some other traumatic event	25	32.9

**Table 5 children-10-01724-t005:** Illness-promoting psychological processes that are a common target of psychological treatment interventions when working with children with FND and their families.

Psychological Process	Explanation/Example of the Psychological Process	Number (%) (*n* = 76)
**Attentional processes**		
Attention to symptoms	Attention to symptoms—by the child or the parents—worsens FND symptoms and amplifies pain.	61 (80.3%)
**Cognitive processes and expectations (a subset relates to symptoms and treatment, and a subset to the self or other situations)**		
Negative/catastrophic ^a^-symptom expectations	Thoughts about the symptom lead to a negative or catastrophic outcome. For example, in response to a muscle spasm and a new pattern of pain in the back—in a boy with motor weakness in the legs and left arm—thinking over and over again, “What is wrong now? How bad is it going to get?” coupled with a visual image of his body being completely paralyzed.	40 (52.6%)
Low sense of control (predictability) over symptoms	Thoughts—coupled with feelings of helplessness—that highlight the child perceived lack of control. For example, “It just happens out of the blue.” or “There is nothing I can do.”	44 (57.9%)
Negative expectations pertaining to treatment interventions	Negative expectations (nocebo effect) are set up in the context of catastrophic-symptom expectations and thoughts/feelings associated with a low sense of control. Negative expectations can undermine the efficacy of strategies that the child needs to learn to manage the symptoms.	30 (39.5%)
Low sense of control (and predictability) with regard to events or expectations in the home or school setting	Thoughts that underline the child’s perceived lack of control. For example, “It’s too hard. I can’t do it. I hate school. They all think I am stupid.”—coupled with feelings of helplessness.	26 (34.2%)
Catastrophic thinking (pertaining to self [symptoms excluded])	“If I don’t get everything right now (in the upcoming test), I’ll never be able to become a surgeon.”	33 (43.4%)
Catastrophic thinking (pertaining to non-self)	For example, ruminating thoughts about the ecological stress that the earth is under, as in “When the glacier melts, we shall have no water, and we shall all die.”	27 (35.5%)
Perfectionistic thinking	For example, “I did not reply to my brother’s text because I could not get my text right.” or “I only got 98% in the math exam” (associated with a feeling of disappointment and sadness).	27 (35.5%)
Self-critical rumination ^b^	Beating oneself up over what one could or should have done. For example, when a child with FND says, “I haven’t tried hard enough. It is my fault I am in hospital. I should have tried harder and done better.” or “The money my parents are spending on my treatment means my family is missing out.”	42 (55.3%)
Obsessive thinking	Unable to switch thought processes away from a certain idea or worry. For example, an adolescent girl with FND was obsessed with the accuracy of her diagnosis. Each morning at ward rounds she grilled her team about her diagnosis, including the pros and cons of formally including other diagnoses on her chart: fibromyalgia, postural orthostatic tachycardia syndrome, complex/chronic pain, irritable bowel syndrome, and so on.	34 (44.7%)
An exclusively negative focus regarding the future, coupled with the inability to be in the present and to celebrate progress in the here and now	For example, an adolescent girl who had presented with leg paralysis, cognitive impairment, and functional seizures complained that everything was getting worse, that her occasional drop attacks made life impossible. She forgot to mention—or celebrate—that she was now dancing around the house and that her cognitive capacities had returned.	30 (39.5%)
**Feeling-related processes (especially feelings that are unacceptable to the self or in the family system)**		
Feeling overly responsible	For example, feeling compelled to act as a confidant and mediator in the school setting to settle distress felt by others or to sort out disputes between friends. Or taking responsibility to keep younger siblings safe while an older sibling is out of control or while parents engage in conflict.	25 (32.9%)
Feeling worried	For example, chronic worries about schoolwork, friendships, or parental wellbeing.	64 (84.2%)
Feeling sad	For example, experiencing sadness or a low mood but not being able to admit and share these feelings with attachment figures.	58 (76.3%)
Feeling anger	For example, experiencing anger but not being able to admit and share feelings of anger.	39 (51.3%)
Feeling guilt	For example, feeling guilt about asking for help or for taking up a hospital bed, or about feeling sad or worried.	26 (34.2%)
Feeling helpless/hopeless about the situation	Nothing will work. No point trying. And so on.	30 (39.5%)
**Avoidance processes**		
Pushing difficult thoughts out of mind (cognition avoidance)	Attempting to manage worries and difficult thoughts—and the associated emotions—by pushing them out of mind. For example, an adolescent girl did not tick the questionnaire item pertaining to family conflict. When asked why, she explained that her father—with whom she was in sharp conflict—was no longer part of her family.	44 (57.9%)
Pushing difficult feelings out of mind (feeling avoidance)	For example (and most commonly), pushing anger out of mind.	37 (48.7%)
Avoidance of activities	For example, avoidance of exercise because it exacerbates pain and can trigger autonomic system activation (such as a panic attack).	38 (50.0%)
**Feeling-related processes related to body state (feeling homeostatic emotions)**		
Disconnecting and the inability to track body states (homeostatic emotions)	Inability to track any change in body state marking increased arousal or distress (e.g., respiratory rate, butterflies in the stomach, changes in tension). For example, an adolescent with leg weakness and panic attacks repetitively stated that she loved school and that school was not stressful despite a relapse of symptoms every time reintegration to school was attempted.	43 (56.6%)
**Attachment-related processes and behavioural processes**		
Being unable to tell parents that not all is well.	For example, not being able to tell parents about feelings of sadness or anger, or about the experience of being worried, overwhelmed.	27 (35.5%)
Not telling parents about what is happening to the child to protect (not burden) parents.	For example, not telling parents about bullying, and trying to manage it all by oneself (but being unable to) in an effort to protect parents from becoming too stressed.	22 (28.9%)
Not being able to ask for help.	For example, from the teacher, thereby perpetuating problems at school.	25 (32.9%)
Amplify signals of distress to activate caregiving behaviour from parents (others)	For example, a boy with an abnormal gait coupled with back pain who signals his distress via loud, lingering wails as he laboriously makes his way to the hospital schoolroom.	32 (42.1%)
**Unresolved loss, unresolved trauma, and unresolved bad experiences**		
Intrusive thoughts/feelings/memories of the adverse event	For example, FND symptoms triggered on the anniversary of a parent’s death or past hospitalization, or when memories of an unresolved trauma are brought to mind in some other way.	38 (50.0%)

^a^ To catastrophize means to “imagine the worst possible outcome of an action or event: to think about a situation or event as being a catastrophe or having a potentially catastrophic outcome” (Merriam-Webster’s online dictionary); ^b^ rumination (as a medical term for a cognitive process) involves “obsessive thinking about an idea, situation, or choice, especially when it interferes with normal mental functioning[,] *specifically*: a focusing of one’s attention on negative or distressing thoughts or feelings” (Merriam-Webster’s online dictionary). For the original version of this table, see Kozlowska and colleagues (2021) [78]. In the current version avoidance processes are listed within their own cluster (avoidance processes). Reproduced with permission © Kasia Kozlowska and Blanche Savage 2021.

**Table 6 children-10-01724-t006:** The three waves of cognitive behavioural therapy.

**Wave 1: Behaviour Therapy**
In the first wave, behaviour therapy methods focus on changing overt behaviour by observing, predicting, and modifying behaviour to promote health and wellbeing. Behaviour therapy involves learning through association and utilizing reinforcement and punishment to modify behaviours. This wave is based on the work of Ivan Pavlov, Burrhus Frederic Skinner, and John Watson.
**Wave 2: Classic CBT**
The second wave of CBT—based on the work of Albert Ellis and Aaron Beck—focuses on the top-down link between maladaptive cognitions and behaviours; the goal is to detect and alter these existing maladaptive patterns and to develop more adaptive ones by identifying, labelling, and reframing cognitive distortions. This wave of CBT also acknowledges the role of behaviour in reinforcing cognitions and feelings, and incorporates bottom-up techniques such as exposure and habit reversal.
**Wave 3: Acceptance CBT**
The third wave of CBT is focused on the person’s relationship to thought and emotion more than the content itself. It emphasizes mindfulness (beginning with the work of Jon Kabat-Zinn), emotions, acceptance, values, and meta-cognition. This wave involves top-down, mindfulness-based, emotion-regulation strategies in which the child utilizes intentional efforts to increase attention and awareness capacities for better control of thoughts and feelings. The objective in third-wave CBT is to help the individual learn to live with painful or unpleasant sensations and with pain in the world, and to accept how things are instead of suffering by trying to change them.
**CBT for FND**
Each of the CBT-based interventions for FND utilizes different techniques selected from the above three waves. For example, retraining and control therapy (ReACT) uses bottom-up strategies, such as principles of habit reversal and mindfulness, to develop opposing responses to FS symptoms, and it challenges catastrophic symptoms expectations [67]. Children are asked to attend to their immediate experience (e.g., what they see and hear, and their physical sensations) immediately prior to the onset of an FS episode, and then to remain aware and conscious of their current experience while engaging in their opposing responses to prevent or stop FS symptoms. Other interventions [63,91] use bottom-up regulation strategies (e.g., slow-breathing techniques, heart rate variability biofeedback, and grounding techniques [similar to those described for retraining and control therapy] [19]) to increase capacity for neuroregulation before implementing other CBT strategies to target specific symptoms.

CBT, cognitive behavioural therapy; FND, functional neurological disorder; FS, functional seizures. Reproduced from Vassilopoulos and colleagues (2022) with permission [92]. © Kasia Kozlowska, Areti Vassilopoulos, and Aaron D. Fobian 2021.

## Data Availability

Consent to place the data in a public repository was not obtained from the children and families who participated in this study.

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
