# Peer review of "Illness-Promoting Psychological Processes in Children and Adolescents with Functional Neurological Disorder"

_children, 2023, doi:10.3390/children10111724_

Round 1
Reviewer 1 Report
The authors reported a review on Illness-Promoting Psychological Processes in Functional Neurological Disorders. The article is well written, and the authors have covered an important topic. I have some comments to the authors:
- The introduction is too long, please try to shorten it.
- A table with the demographic/clinical feature of the patients might help the reader to better follow the manuscript. The authors should also specify the kind of FND involved, for instance how many of the patients with motor symptoms had tremor, dystonia etc.
- Have the authors tried to do some correlations between the clinical phenpype (PNES and FMD for instance) and the questionnaire scores?
- Most of the clinical vignettes might be included in the Suplementary materials
Author Response
Responses to Reviewer 1
The authors reported a review on Illness-Promoting Psychological Processes in Functional Neurological Disorders. The article is well written, and the authors have covered an important topic. I have some comments to the authors:
- The introduction is too long, please try to shorten it.
We have made some cuts to the introduction as requested. Please see tracked changes. We note, however, that as far as we know, there is no published summary of paediatric studies pertaining to cognitive processes in children with FND. In this context, a summary of the available information is important.
- A table with the demographic/clinical feature of the patients might help the reader to better follow the manuscript. The authors should also specify the kind of FND involved, for instance how many of the patients with motor symptoms had tremor, dystonia etc.
Thank you for raising this important and complex question.
Most of the children in the sample (n = 49; 64%) fit the mixed FND category. Mixed FND refers to presentations that are characterised by more than on FND symptom. In our tertiary-care hospital setting, mixed FND is the presentation that we see most frequently. To fill out this information for Reviewer 1, we have added a more detailed legend to Figure 5 (which depicts the type of FND presentations in the cohort). We hope this gives a more nuanced idea of the range of FND symptoms that the children presented with. As the reviewer will see, there is a smattering of everything, with 64% of the children presenting with various combinations of symptoms. We hope the addition of this more detailed legend is helpful.
- Have the authors tried to do some correlations between the clinical phenpype (PNES and FMD for instance) and the questionnaire scores?
Following on from the previous answer, because the majority of our sample was mixed FND—had various combinations of symptoms—we did not run correlations between phenotypes. None of the phenotypes is “clean,” as it were.
Our previous data sets suggest that children who have functional seizures show more pronounced differences in neural network dysregulation (Rai et al. 2022) and a greater increase in resting-state gamma power on graph theory analysis (Radmanesh, 2020). In this context, in response to Reviewer 1’s question, we ran an independent t-test looking at the number of illness-promoting cognitive processes identified in the group with functional seizures and those without. Unsurprisingly, we found no difference (t(74) = -.907; p = .368).
The reality is that the mixed FND nature of our sample—a feature of all our samples—is not conducive to an examination of different clinical phenotypes.
Rai, S., et al. (2022). "Altered resting-state neural networks in children and adolescents with functional neurological disorder." Neuroimage Clin 35: 103110.
Radmanesh, M., et al. (2020). "Activation of Functional Brain Networks in Children and Adolescents with Psychogenic Non-epileptic Seizures." Frontiers in Human Neuroscience 14: 339.
- Most of the clinical vignettes might be included in the Suplementary
Our very strong preference is to keep the vignette material in the main text. The continual feedback from our clinical readers—ordinary clinicians (psychologists, psychiatrists, physiotherapists, occupational therapists, speech therapists)—is that they find our work helpful materials because it always includes a mix of empirical research coupled with clinical examples of what the data look like in clinical practice. For clinicians who may want to work in the field, the research data with no vignettes is meaningless. The clinicians need the specific examples to understand what the data mean in terms of daily clinical practice. They need the specific examples to see the texture of the material. Whilst paediatricians and neurologists do not need this texture—because they do not engage in detailed psychotherapeutic work—clinicians do need this texture in order to provide treatment.
Since the recurring feedback from clinicians is the utility of the clinical elements of our papers, we think it important that this material is left in the main text.
On a final note, neurologists and paediatricians always lament the fact that psychologists do not want to take on their patients with FND. One of the reasons for this state of affairs is that many psychologists do not do not know what to do, and one of the reasons for this lack of knowledge base is that the literature has become so cryptic (due to short word limits) that there is little detailed information as to what to do. For example, there is a wonderful study by Fobian and colleagues that describes outcomes of retraining and control therapy (ReACT) for functional seizures. But the word limits of the article mean that the actual treatment is not described in any detail. The consequence is that clinicians reading the material are not given sufficient information to “know what to do”. It is for this reason that we have tried wherever possible to include clinical information alongside research data. The data in isolation are not helpful on the front line in the clinic. And the journal Children, because of its more generous word limit, currently has an edge over other journals whose strict word limits do not allow for a discussion of how research results are translated into clinical practice.
It is also important to note that Reviewer 2 specifically noted that the clinical content was very important, “I believe the clinical content is very helpful for the readers to learn about and use as it rests on sound theories and literature”.
Fobian, A. D., et al. (2020). "Retraining and control therapy for pediatric psychogenic non-epileptic seizures." Ann Clin Transl Neurol.
Reviewer 2 Report
Thank you for allowing me to review the manuscript "Illness-promoting psychological processes in children and adolescents with functional neurological disorder." Given the rise in cases that are now being seen routinely in clinics around the globe this paper is of interest to the readers of Children. I have some major concerns and then will share some minor edits.
Major Concern:
This report is primarily a clinical report for the treatment of FND vs as presented as the small study where the authors look at a control cohort for 3 self reported surveys that don't pertain to most of this manuscript. I believe it has been previously established that kids with FND experience more distress and ACES. I am not sure what this part of the manuscript adds. It would have been informative if the controls also got clinically assessed and Table 4 & 5 documented results of both kids with FND and controls.
This paper should instead be presented as a clinical report based on existing literature and the experience of the authors treating patients in the inpatient setting based on the clinical interview and surveys completed during the study. I believe the clinical content is very helpful for the readers to learn about and use as it rests on sound theories and literature.
Discussion section overly generalizes and extends the finds of this study. The conclusions are made looking at the clinical cohort and their clinical data and trajectory of treatment vs compared to the control and therefore need to be more specific to this sample. Some very broad generalizations have been made in the discussion section that need to be removed.
Minor edits:
Page 14: column 1 edit No being able to = Not being able to
Page 27: 7.2 3rd sentence should be triggered "by" her father's near...
Page 28: 8.2 heading. Edit to "using hypnosis to imagine (not image) the past
Page 29: 9 First sentence
Given (not Give). Same section: 2nd paragraph last sentence. That individual work which a child fails.
I do believe streamlining this paper's discussion and focusing on the clinical data would still be an important contribution and interesting to the readers of Children.
Author Response
Thank you for allowing me to review the manuscript "Illness-promoting psychological processes in children and adolescents with functional neurological disorder." Given the rise in cases that are now being seen routinely in clinics around the globe this paper is of interest to the readers of Children. I have some major concerns and then will share some minor edits.
Major Concern:
This report is primarily a clinical report for the treatment of FND vs as presented as the small study where the authors look at a control cohort for 3 self reported surveys that don't pertain to most of this manuscript. I believe it has been previously established that kids with FND experience more distress and ACES. I am not sure what this part of the manuscript adds. It would have been informative if the controls also got clinically assessed and Table 4 & 5 documented results of both kids with FND and controls.
Thank you for raising this important point. There are two reasons for including the questionnaire and GAF data pertaining to controls for this sample.
First, when we have not included any comparison data for the ACEs that the children and families have reported, we have been severely criticised for claiming (with no evidence) that this group of children has higher ACEs. That is, those criticizing us have raised the question, “how do you know this rate of ACEs is different to those experienced by other children?” The learning point for us has been that if we do have comparison data, we need to include it. A
Second, Reviewer’s 2 understanding “that it has been previously established that kids with FND experience more distress and ACEs” is not entirely correct. For example, our colleagues in Alabama USA have repeatedly mentioned to us that their patient cohorts do not report high levels of ACEs and that they are not characterised by high rates of comorbid anxiety and depression. Unlike us, however, our colleagues see a less distressed and disabled group of children with functional seizures in an outpatient setting. In this context, we think it is very important to establish the clinical characteristics of our sample. Our tertiary-care hospital sample was made up of very unwell children with FND. They had high rates of ACEs, functional impairment, and mental health disorders. Importantly, however, not all paediatric FND samples have the same levels of distress, ACEs, and mental health psychopathology.
In sum, we think it very important to retain this information, which provides clear information that our cohort of children is on the more severe end of the spectrum.
This paper should instead be presented as a clinical report based on existing literature and the experience of the authors treating patients in the inpatient setting based on the clinical interview and surveys completed during the study. I believe the clinical content is very helpful for the readers to learn about and use as it rests on sound theories and literature.
Thank you for raising this question because the methodology of the study is unique.
The children in this study agreed to participate in a research study called, “Stress system activation in Australian children and adolescents with Functional Neurological Disorder”. The study had multiple components (collection of clinical data, cortisol study, resting-state MRI, resting-state spectroscopy, and so on). As part of this clinical data component, we prospectively documented the psychological processes that emerged in the therapeutic work with the child during a two-week mind-body rehabilitation program. The information gathered emerged in the context of therapeutic work with the child. It was discussed with the child, discussed in family sessions, and was included in the child’s report. It became a focus of therapeutic work, both during the program and in subsequent psychotherapy.
The term “clinical report” fails to capture the prospective methodology that we used. Likewise, “clinical interview” does not capture the interactions within the therapeutic space. In the therapeutic space, we were not interviewing the child but working collaboratively with the child. Identifying the processes was a key goal of the admission (as it always is) and a collaborative effort.
In sum, it is not accurate to say that the information documented here emerged via clinical interview. It did not. It emerged within the therapeutic conversations and explorations of the mind-body admission—the psychotherapeutic processes. It certainly did not emerge via surveys. The data was prospectively collected by documenting the therapeutic content of psychotherapy sessions and therapeutic interactions during the admission.
As noted by Reviewer 2, the vignettes do reflect our clinical experience. They are our effort to communicate in a more nuanced way what the data mean for daily clinical practice.
We have had a look at the manuscript and tried to make this methodology more clear. Please see the tracked changes.
Discussion section overly generalizes and extends the finds of this study. The conclusions are made looking at the clinical cohort and their clinical data and trajectory of treatment vs compared to the control and therefore need to be more specific to this sample. Some very broad generalizations have been made in the discussion section that need to be removed.
We have tried to make it more clear for the reader that the discussion pertained to the clinical cohort only.
We also took out the paragraph that did not directly relate to the study (i.e., that overgeneralised).
Minor edits:
Page 14: column 1 edit No being able to = Not being able to
Thank you. The typo is fixed.
Page 27: 7.2 3rd sentence should be triggered "by" her father's near...
Thank you. The typo is fixed.
Page 28: 8.2 heading. Edit to "using hypnosis to imagine (not image) the past
Thank you. The typo is fixed.
Page 29: 9 First sentence
Given (not Give). Same section: 2nd paragraph last sentence. That individual work which a child fails.
Thank you. The first typo has been fixed and the second already seemed to be fixed in the version downloaded from the journal site.
I do believe streamlining this paper's discussion and focusing on the clinical data would still be an important contribution and interesting to the readers of Children.
We thank Reviewer 2 for his/her feedback. Please see changes to manuscript via the track-change function.
Round 2
Reviewer 2 Report
Thank you for allowing me to review this revision which the authors have written based on the feedback received. While I do believe the manuscript is clearer, I think it would be important to be clear in the paper that the majority of the data presented was based on the clinical sample and only 3 measures have a comparative control group. Throughout the paper and particularly in the discussion, please start with a clarifying sentence that the majority of the findings and discussions are based on the clinical experience/clinical surveys with this cohort in an inpatient rehab setting (where one would assume youth with more severe and refractory FNSD symptoms are seen). Your point about ACES is well taken but that too should not be generalized to all youth with FNSD. As you are aware, close to 90% of patients seen by neurologists in the acute inpatient/outpatient setting are diagnosed with FND and most recover very quickly. The assumption that those patients would also have a higher level of ACES can not be made from this study. Please specify throughout the paper that these findings pertain to a subset of patients as you mentioned who are sicker and therefore needing a higher level of intervention and the conclusions only pertain to that subset of youth with FND.
Overall, I do think that we need to tighten this paper more. Be clear about what aspects have a controlled control sample and which aspects don't-starting with the abstract. Further throughout the discussion, we need to base conclusions focused on patients needing a higher level of intervention and therefore inherently with greater co-morbid psychiatric conditions, significantly compromised function and higher ACES.
Author Response
Reviewer 2: Second round of Revision Comments
We thank Reviewer 2 for his/her comments. We are in perfect agreement that being clear about the characteristics of our sample is really important. We work in a tertiary care hospital for the state of NSW, Australia. And in this context, we do see the most sick patients from around NSW. As noted by Reviewer 2 our sample also has high psychiatric comorbidity. And of course, this is not true of all samples. In this context, a clear description of our sample is really important.
We have made further amendments to the abstract and text to make the manuscript even more clear.
The one interesting point is that our children and adolescents with FND, have very severe symptoms, but their symptoms are not refractory. Because they receive intensive treatment early, the majority of them recover. We realise this is very different to adult samples where refractory FND is much more common.
Thank you for allowing me to review this revision which the authors have written based on the feedback received. While I do believe the manuscript is clearer, I think it would be important to be clear in the paper that the majority of the data presented was based on the clinical sample and only 3 measures have a comparative control group.
Thank you. This has been attended to. See track changes.
Throughout the paper and particularly in the discussion, please start with a clarifying sentence that the majority of the findings and discussions are based on the clinical experience/clinical surveys with this cohort in an inpatient rehab setting (where one would assume youth with more severe and refractory FNSD symptoms are seen).
Thank you. This has also been attended to. See track changes.
Your point about ACES is well taken but that too should not be generalized to all youth with FNSD. As you are aware, close to 90% of patients seen by neurologists in the acute inpatient/outpatient setting are diagnosed with FND and most recover very quickly. The assumption that those patients would also have a higher level of ACES can not be made from this study. Please specify throughout the paper that these findings pertain to a subset of patients as you mentioned who are sicker and therefore needing a higher level of intervention and the conclusions only pertain to that subset of youth with FND.
Thank you. We agree entirely that being clear about the characteristics of the current patient cohort is important. We have added a line to the limitations section pertaining to generatability (or rather not).
Overall, I do think that we need to tighten this paper more. Be clear about what aspects have a controlled control sample and which aspects don't-starting with the abstract.
Thank you. This has also been attended to. See track changes.
Further throughout the discussion, we need to base conclusions focused on patients needing a higher level of intervention and therefore inherently with greater co-morbid psychiatric conditions, significantly compromised function and higher ACES.
Thank you. This has also been attended to. See track changes.